# Human-Aligned Image Models Improve Visual Decoding from the Brain

**Nona Rajabi**[1]  **Antônio H. Ribeiro**[2*]  **Miguel Vasco**[1*]  **Farzaneh Taleb**[1]  **Mårten Björkman**[1]  **Danica Kragic**[1]

## Abstract

Decoding visual images from brain activity has significant potential for advancing brain-computer interaction and enhancing the understanding of human perception. Recent approaches align the representation spaces of images and brain activity to enable visual decoding. In this paper, we introduce the use of human-aligned image encoders to map brain signals to images. We hypothesize that these models more effectively capture perceptual attributes associated with the rapid visual stimuli presentations commonly used in visual brain data recording experiments. Our empirical results support this hypothesis, demonstrating that this simple modification improves image retrieval accuracy by up to 21% compared to state-of-the-art methods. Comprehensive experiments confirm consistent performance improvements across diverse EEG architectures, image encoders, alignment methods, participants, and brain imaging modalities[1].

## 1. Introduction

Decoding visual information from brain activity has long been a key goal in neuroscience and computer vision, with the potential to revolutionize brain-computer interfaces, deepen our understanding of neural processes, and enable advanced assistive technologies. Current state-of-the-art methods for visual decoding rely on three core components: *(i)* a *brain-signal encoder* that processes high-dimensional, multivariate time-series data from the brain; *(ii)* a pre-trained *image encoder* capable of generating semantically rich image embeddings; and *(iii)* a *self-supervised loss function* that aligns brain signals and image embeddings within a

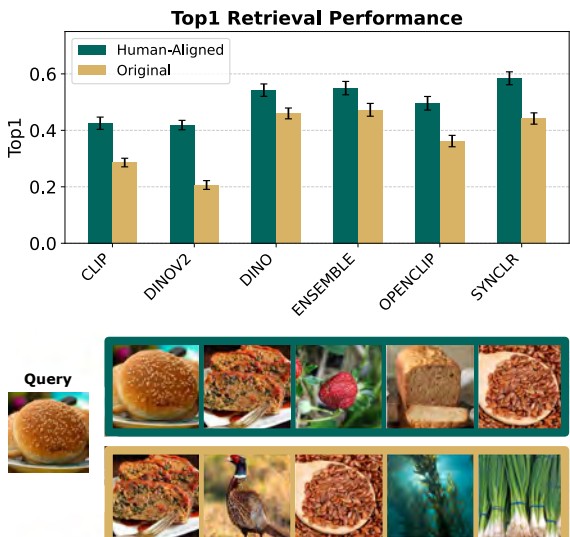

**Top1 Retrieval Performance**

*Figure 1.* **Performance of image retrieval from EEG data**. We show how the use of human-aligned versions of standard image encoders consistently leads to an increase in performance across image retrieval tasks from brain activity. The bars represent the top-1 retrieval performance averaged over subjects.

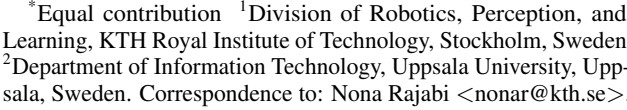

shared representation space.

Despite progress, visual decoding, especially from low signal-to-noise ratio data like electroencephalogram (EEG), remains a significant challenge. Previous approaches have explored various brain signal encoders (Benchetrit et al., 2023; Song et al., 2024; Li et al., 2024), additional encoding models (Scotti et al., 2023; Li et al., 2024), and auxiliary loss functions (Ferrante et al., 2024). Yet, overall performance remains limited; for example, state-of-the-art EEG-based models achieve only 28% top-1 retrieval accuracy from a database of 200 unseen test images. Here, we take another approach and focus on the nature of the image encoder and how it affects the performance of visual decoding methods.

We investigate the use of *human-aligned* image representation models (Fel et al., 2022; Muttenthaler et al., 2023b; Sundaram et al., 2024) for visual decoding from brain signals. These models are specifically designed to ensure that images perceived as similar by humans are closer in the representation space, effectively aligning the embeddings with human perception. This alignment is particularly rele-

---

*Equal contribution  [1]Division of Robotics, Perception, and Learning, KTH Royal Institute of Technology, Stockholm, Sweden  [2]Department of Information Technology, Uppsala University, Uppsala, Sweden. Correspondence to: Nona Rajabi <nonar@kth.se>.

*Proceedings of the 42nd International Conference on Machine Learning*, Vancouver, Canada. PMLR 267, 2025. Copyright 2025 by the author(s).

[1]All codes are available at https://github.com/NonaRjb/AlignVis.git

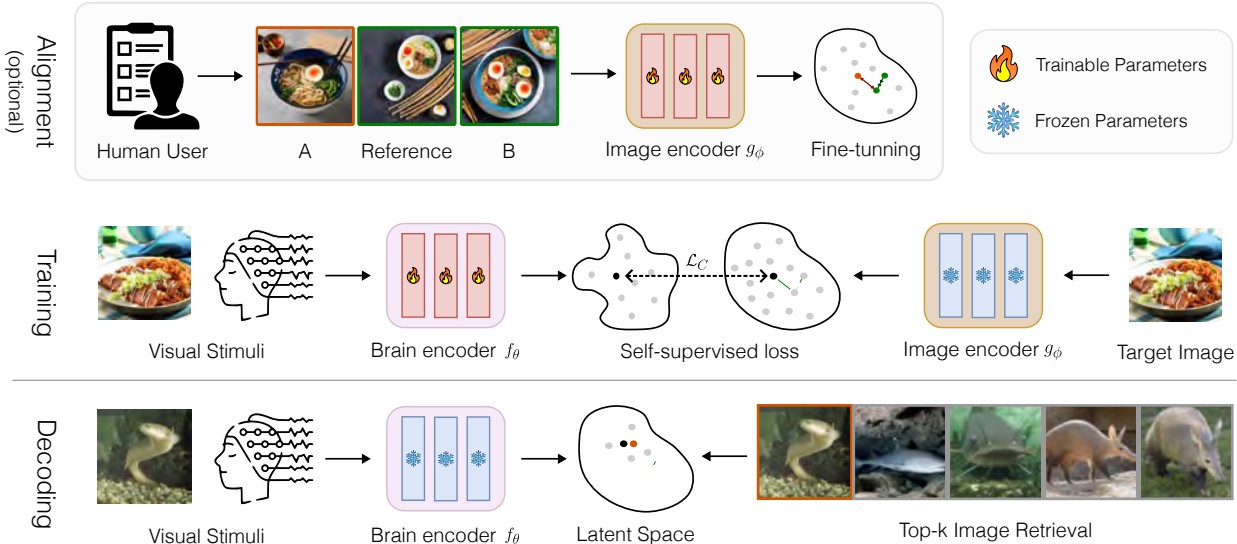

Figure 2. **Learning to map brain activity to images**. Image embeddings $\mathbf{v} \in \mathbb{R}^d$ are extracted from frozen image encoders $f_\theta(.)$, while brain signal embeddings $\mathbf{w} \in \mathbb{R}^d$ are computed using modality-specific encoders $g_\phi(.)$. A contrastive loss then aligns brain-signal representations to the image space. We consider both original pre-trained image encoders and human-aligned versions fine-tuned on human image similarity judgments. Images for the similarity judgment dataset and retrieval results are borrowed from *NIGHTS* (Fu et al., 2023) and *THINGS* (Hebart et al., 2019) datasets.

vant for decoding visual information from brain signals, as human judgments emerge from underlying brain processes.

We demonstrate that incorporating pre-trained, human-aligned image encoders significantly improves visual decoding performance from brain signals. Through an extensive empirical study spanning multiple brain modalities, brain-signal encoders, and image foundation models, we show consistent and substantial improvements achieved through human alignment. Furthermore, we analyze the differences between models trained with human-aligned and baseline image encoders, revealing that human-aligned models focus on signal components known to be relevant to visual processing. By leveraging human-aligned image encoders, we improve the accuracy of state-of-the-art models by up to 21%, establishing a new direction for visual decoding from the brain (Figure 1).

Our key contributions are as follows:

- We propose and demonstrate that *human-aligned models consistently and significantly improve visual retrieval from the brain*.

- We conduct an *extensive empirical study* of various human-alignment methods, image and brain encoders, and neuroimaging modalities.

- We hypothesize that human *similarity judgment datasets capture early human impressions* of images and are more closely aligned with the brain signals. To support this hypothesis, we present experiments that provide further evidence.

## 2. Brain-to-Image Decoding via Latent Space Alignment

We frame the problem of visual decoding from the brain as retrieving the corresponding viewed image from a given image database based on brain activity. We exploit an architecture composed of a brain-signal encoder and an image encoder. After training the brain encoder, the test brain activity is embedded, and its distances to all images in the test database are computed. The images are then ranked by their distance to the brain embedding, and the top-$k$ images are retrieved. The procedure and overall architecture of the method are summarized in Figure 2.

Formally, let $\mathbf{x}$ denote an image from a subset of an image dataset, and let $\mathbf{b}$ represent the brain-activity measurement recorded from a subject while observing $\mathbf{x}$. We define two neural network-parameterized functions, $f_\theta$ and $g_\phi$, that map images and brain signals, respectively, to a shared latent vector space $Z \in \mathbb{R}^d$:

$$\mathbf{x} \mapsto f_\theta(\mathbf{x}) = \mathbf{v}, \quad \mathbf{b} \mapsto g_\phi(\mathbf{b}) = \mathbf{w}, \quad \mathbf{v}, \mathbf{w} \in \mathbb{R}^d. \quad (1)$$

Our goal is to train the brain encoder $g_\phi$ such that the embeddings $(\mathbf{v}, \mathbf{w})$ for each pair $(\mathbf{x}, \mathbf{b})$ are *aligned* in the shared latent space $Z \in \mathbb{R}^d$. To achieve this, we use frozen image encoders $f_\theta$ derived from pre-trained image or image-text models, which provide rich, generalizable representations without requiring fine-tuning. Aligning brain embeddings $\mathbf{w}$ with the image embeddings $\mathbf{v}$ enables the retrieval of the corresponding viewed image $\mathbf{x}$ from the brain signal.

This alignment is enforced using the multimodal InfoNCE loss (Radford et al., 2021):

$$\mathcal{L}_C = -\frac{1}{N} \sum_{i=1}^{N} \left[ \log \frac{\exp(\text{sim}(\mathbf{w}_i, \mathbf{v}_i)/\tau)}{\sum_{j=1}^{N} \exp(\text{sim}(\mathbf{w}_i, \mathbf{v}_j)/\tau)} \right.$$
$$\left. + \log \frac{\exp(\text{sim}(\mathbf{v}_i, \mathbf{w}_i)/\tau)}{\sum_{j=1}^{N} \exp(\text{sim}(\mathbf{v}_i, \mathbf{w}_j)/\tau)} \right], \quad (2)$$

where $\tau$ is the temperature parameter and $\text{sim}(a,b)$ is a distance metric between the two embeddings: in our case, we employ cosine similarity. The same subscripts denote matching pairs. We investigate the impact of image encoders on visual decoding from brain activity, specifically assessing whether human perceptual biases in image representation spaces influence performance. Our hypothesis is:

> **Hypothesis**: Replacing the image encoder $f_\theta(\mathbf{x})$ with its perceptually aligned version significantly improves visual decoding performance from brain signals.

We based our hypothesis on the intuition that human similarity judgments originate from brain processes and thus reflect the similarities between brain signals corresponding to image stimuli. Additionally, we propose that this improvement is linked to the experimental design of the image-similarity dataset. Specifically, since brain signals are elicited by a rapid visual stimulus (100 ms viewing time), they primarily capture early impressions of images, such as shape, orientation, and color. These features are better represented in image-similarity datasets focused on early perceptual attributes. We corroborate this hypothesis through the experiments provided in later sections.

## 3. Related Work

Decoding images from brain activity and aligning vision models with human perception are two independently researched directions. However, perceptually aligned models have yet to be explored for visual decoding from the brain. This section reviews prior work in both areas, highlighting the gap our study addresses.

### 3.1. Decoding Images from Brain Activity

Understanding how humans perceive images and decode this information from brain signals has been a long-standing question in neuroscience and, more recently, in computer vision (Hubel & Wiesel, 1962; Kamitani & Tong, 2005; Scotti et al., 2024). Early studies focused on decoding simple visual features like stimulus orientation (Kamitani & Tong, 2005) and basic geometric shapes (Du et al., 2017). Advances in computer vision (Radford et al., 2021; Dosovitskiy, 2020; Rombach et al., 2022) and large-scale datasets (Gifford et al., 2022; Allen et al., 2022; Hebart et al., 2023) have enabled the retrieval and reconstruction of complex images from brain activity. Research has explored natural image and video decoding using functional magnetic resonance imaging (fMRI) (Scotti et al., 2023; 2024), magnetoencephalogram (MEG) (Benchetrit et al., 2023), and EEG (Song et al., 2024; Li et al., 2024). While fMRI achieves higher decoding accuracy, MEG and EEG offer better temporal resolution and accessibility. A common approach in these studies maps brain signals into pre-trained image representation spaces like CLIP (Radford et al., 2021) and DINO (Caron et al., 2021) using a self-supervised loss, such as InfoNCE (Equation 2). These mappings enable image retrieval and can serve as conditioning signals for image generative models like Stable Diffusion (Rombach et al., 2022) to reconstruct viewed images. Unlike prior work focusing on brain encoders and auxiliary modalities like text, we investigate the role of image encoders. Specifically, we examine how aligning image encoders with human perception impacts visual decoding from brain signals.

### 3.2. Perceptual Alignment of Vision Models

Early computer vision models were inspired by human perception (Fukushima, 1980; Serre et al., 2007; Krizhevsky et al., 2012), but modern models often diverge from it (Golan et al., 2020; Lindsay, 2021; Kumar et al., 2022; Fel et al., 2022; Muttenthaler et al., 2023a). Recent research has sought to bridge this gap by aligning machine vision models with human perception. Fel et al. (2022) proposed a neural harmonizer that co-trains deep networks to align with human visual strategies while maintaining task accuracy. Their approach leverages datasets with human judgments on salient image features from ImageNet (Russakovsky et al., 2015). Similarly, Muttenthaler et al. (2023b) introduced *gLocal*, a linear transformation that aligns the global structure of neural representations with human similarity judgments while preserving local structure. Unlike naive transformations that degrade downstream task performance, *gLocal* maintains or slightly improves performance while enhancing alignment with human perception. More recently, Fu et al. (2023) introduced *Dreamsim*, a metric for fine-tuning vision models on a human image similarity dataset. Sundaram et al. (2024) demonstrated that *Dreamsim*-aligned models outperform unaligned ones on several downstream vision tasks, including object counting, instance retrieval, and image segmentation.

## 4. Materials and Methods

Here we describe the datasets, pre-trained image encoders, and EEG encoder architectures employed in our study.

### 4.1. Datasets

We used the *Things EEG2* dataset (Gifford et al., 2022) to train and evaluate our framework. *Things EEG2* is, to our knowledge, the only dataset with a sufficient number of participants and trials per participant to support more

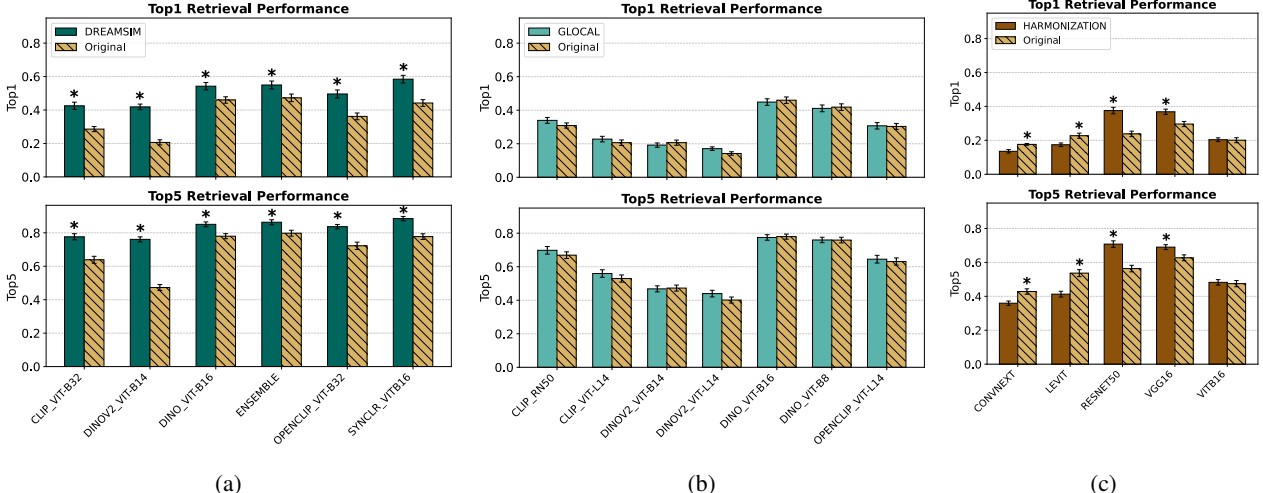

*Figure 3.* **200-way image retrieval performance from EEG signals**. For this task, we employ the *NICE* EEG encoder (Song et al., 2024) and evaluate different human-alignment methods and image encoders. Each vertical pair of panels demonstrates the results for one human-alignment method. The results highlight a consistent improvement in image decoding performance when considering human-aligned image encoders. Stars represent significant difference confirmed by a paired T-test ($p < 0.05$). Best viewed with zoom.

robust and generalizable findings. It also employs a random-order design, avoiding biases associated with block-order designs (Li et al., 2020).

EEG data were recorded from 10 participants exposed to visual stimuli via a rapid serial visual presentation (RSVP) paradigm (100 ms per image). The training set includes 1,654 unique concepts, each with 10 images shown in random order and repeated 4 times, totaling $1654 \times 10 \times 4$ samples per participant. The test set contains 200 distinct concepts, each with 1 image shown 80 times, yielding $200 \times 1 \times 80$ samples per participant. Training and test concepts are disjoint. To improve the signal-to-noise ratio, repetitions were averaged, reducing training and test samples to $1654 \times 10$ and $200 \times 1$ per participant. Models were trained with a 90/10% split and evaluated on the test set.

For the results in Section 5.4, we used preprocessed MEG data from (Hebart et al., 2023), recorded from 4 participants in response to 1,854 unique concepts (500 ms per image). The training set contains $1854 \times 12 \times 1$ samples, while the test set includes $200 \times 1 \times 12$ samples per participant. We excluded the 200 test concepts from training and averaged the 12 repetitions per test concept, resulting in $1654 \times 12$ training samples and $200 \times 1$ test samples per participant. For both EEG and MEG, models were trained on signals from 0 to 1000 ms relative to stimulus onset. Further dataset details are in Appendix A.

### 4.2. Pretrained Vision Models

We train the whole EEG encoder by aligning its final layer representations to a pretrained vision model's representations. For this purpose, we use image embeddings from

vision models aligned to human perceptual judgments proposed by Sundaram et al. (2024) (which we refer to as *Dreamsim*) and compare them with their original unaligned version. Specifically, we use vision transformer (ViT) (Dosovitskiy, 2020) backbones including *CLIP* (Radford et al., 2021), *OpenCLIP* (Cherti et al., 2023), *DINO* (Caron et al., 2021), *DINOv2* (Oquab et al., 2024), and *SynCLR* (Tian et al., 2024). We also include results obtained by concatenated features from *DINO*, *CLIP*, and *OpenCLIP* referred to as the *Ensemble* model (Sundaram et al., 2024).

### 4.3. EEG Architectures and Training

We used the *NICE* EEG encoder proposed by Song et al. (2024) to embed EEG signals in the experiments described in Sections 5.1, 5.3, and 5.4. *NICE* is a recent convolutional neural network (CNN) that applies temporal and spatial convolutions to multivariate time-series input data. Additionally, in Section 5.2, we evaluated our method with other EEG encoders by training three recent architectures: *EEG-Netv4* (Lawhern et al., 2018), a widely used CNN for EEG tasks; *EEGConformer* (Song et al., 2022), a transformer-based model designed for EEG signals; and *ATM-S* (Li et al., 2024), which combines a transformer with a CNN.

To speed up the training, we pre-computed the image embeddings using the frozen image encoders. *Dreamsim* models were used as human-aligned image encoders throughout the paper unless explicitly mentioned otherwise. We trained all models using the InfoNCE loss (Equation 2) with five different seeds (which also resulted in five different train/val splits). All results presented were averaged over all seeds. More training details are presented in the Appendix B.

*Table 1.* **200-way top-1 image retrieval performance from EEG signals.** We consider different EEG and image encoders during the training of the models. The results show that using human-aligned (HA) image encoders consistently improves the performance of the overall model, regardless of the specific image and EEG encoder employed. Best viewed with zoom.

| | Ensemble | | SynCLR | | CLIP | | OpenCLIP | | DINO | | DINOv2 | |
|---|---|---|---|---|---|---|---|---|---|---|---|---|
| | HA | Base | HA | Base | HA | Base | HA | Base | HA | Base | HA | Base |
| EEGNetV4 | **39.51 ± 1.42** | 33.94 ± 1.80 | **38.98 ± 1.27** | 31.52 ± 1.64 | **28.03 ± 1.40** | 20.55 ± 0.96 | **31.31 ± 1.74** | 24.10 ± 1.49 | **37.46 ± 1.42** | 31.34 ± 1.43 | **27.45 ± 1.12** | 15.93 ± 1.07 |
| EEGConformer | **23.78 ± 1.25** | 19.62 ± 1.13 | **26.25 ± 1.01** | 18.05 ± 1.37 | **19.02 ± 1.01** | 13.59 ± 0.81 | **21.96 ± 0.95** | 14.61 ± 0.89 | **23.92 ± 1.30** | 19.48 ± 0.89 | **18.14 ± 1.04** | 9.26 ± 0.63 |
| NICE | **54.96 ± 2.37** | 47.27 ± 2.27 | **58.44 ± 2.29** | 44.19 ± 2.02 | **42.52 ± 2.17** | 28.59 ± 1.52 | **49.59 ± 2.40** | 36.20 ± 2.02 | **54.24 ± 2.19** | 45.99 ± 1.92 | **41.87 ± 1.67** | 20.64 ± 1.55 |
| ATM-S | **56.36 ± 2.14** | 48.89 ± 1.97 | **62.53 ± 1.59** | 46.15 ± 1.59 | **46.06 ± 1.62** | 29.87 ± 1.41 | **53.25 ± 1.93** | 40.13 ± 1.77 | **56.30 ± 1.92** | 48.04 ± 1.61 | **44.73 ± 1.61** | 22.91 ± 1.49 |

*Table 2.* **Cross-subject 200-way top-1 image retrieval performance from EEG signals.** We encode EEG using the *NICE* encoder and consider different image encoders during the training of the models. HA refers to the human-aligned models. Best viewed with zoom.

| | Ensemble | | SynCLR | | CLIP | | OpenCLIP | | DINO | | DINOv2 | |
|---|---|---|---|---|---|---|---|---|---|---|---|---|
| | HA | Base | HA | Base | HA | Base | HA | Base | HA | Base | HA | Base |
| S1 | 11.4 ± 1.71 | 12.2 ± 1.36 | 9.4 ± 1.16 | 10.0 ± 0.84 | 10.3 ± 1.96 | 6.0 ± 1.05 | 11.5 ± 0.55 | 9.6 ± 0.37 | 10.6 ± 1.11 | 10.0 ± 0.71 | 9.8 ± 1.21 | 9.5 ± 1.00 |
| S2 | 15.0 ± 1.67 | 15 ± 1.48 | 13.1 ± 1.83 | 13.2 ± 1.63 | 12.8 ± 1.12 | 8.5 ± 0.95 | 11.8 ± 2.64 | 10.4 ± 1.85 | 15.1 ± 1.25 | 13.5 ± 1.73 | 13.5 ± 1.38 | 10.8 ± 1.21 |
| S3 | 8.1 ± 0.98 | 9.5 ± 4.07 | 12.8 ± 1.91 | 8.7 ± 1.50 | 11.9 ± 1.98 | 7.5 ± 1.00 | 10.5 ± 3.18 | 7.1 ± 0.73 | 9.1 ± 1.31 | 9.4 ± 1.24 | 8.4 ± 1.88 | 7.8 ± 0.75 |
| S4 | 9.5 ± 1.58 | 12.0 ± 1.05 | 10.2 ± 1.50 | 9.4 ± 1.36 | 9.9 ± 1.24 | 6.8 ± 0.40 | 10.0 ± 2.14 | 6.1 ± 1.24 | 9.3 ± 1.03 | 9.2 ± 0.98 | 9.6 ± 1.32 | 10.8 ± 1.21 |
| S5 | 7.3 ± 2.42 | 7.9 ± 0.86 | 7.6 ± 0.58 | 8.6 ± 1.28 | 7.4 ± 1.36 | 4.4 ± 1.39 | 6.1 ± 1.88 | 6.2 ± 1.29 | 7.8 ± 1.17 | 8.1 ± 1.68 | 7.1 ± 0.97 | 6.4 ± 1.16 |
| S6 | 21.8 ± 1.60 | 18.9 ± 1.24 | 17.0 ± 1.22 | 14.4 ± 1.36 | 16.7 ± 1.47 | 12.0 ± 0.48 | 13.0 ± 2.39 | 10.9 ± 1.39 | 13.0 ± 2.55 | 15.5 ± 1.97 | 11.87 ± 1.78 | 11.3 ± 1.66 |
| S7 | 10.8 ± 1.72 | 10.5 ± 1.45 | 10.7 ± 2.48 | 8.8 ± 1.03 | 8.5 ± 1.52 | 5.9 ± 0.97 | 7.6 ± 1.71 | 7.0 ± 1.38 | 9.5 ± 1.30 | 10.2 ± 1.86 | 9.5 ± 1.30 | 5.5 ± 0.45 |
| S8 | 14.5 ± 1.45 | 13.6 ± 0.86 | 15.9 ± 1.53 | 12.7 ± 0.98 | 12.4 ± 1.56 | 10.5 ± 1.00 | 13.6 ± 0.71 | 10.0 ± 0.55 | 14.2 ± 1.54 | 13.2 ± 1.03 | 13.8 ± 0.93 | 9.6 ± 1.35 |
| S9 | 13.9 ± 2.44 | 13.3 ± 2.18 | 13.8 ± 0.51 | 12.4 ± 1.65 | 10.0 ± 1.45 | 7.6 ± 0.66 | 11.4 ± 1.46 | 9.7 ± 1.91 | 14.2 ± 0.93 | 11.8 ± 1.12 | 10.0 ± 0.71 | 9.8 ± 1.81 |
| S10 | 15.8 ± 2.96 | 16.4 ± 0.73 | 13.3 ± 1.36 | 12.0 ± 1.30 | 14.1 ± 1.20 | 10.2 ± 1.29 | 12.3 ± 1.50 | 9.2 ± 1.94 | 12.6 ± 2.30 | 13.2 ± 1.03 | 10.1 ± 1.65 | 11.9 ± 1.32 |
| Avg. | 12.8 ± 1.37 | 12.9 ± 1.04 | 12.4 ± 0.92 | 11.0 ± 0.68 | **11.4 ± 0.87** | 7.9 ± 0.75 | **10.8 ± 0.74** | 8.6 ± 0.58 | 11.5 ± 0.82 | 11.4 ± 0.75 | 10.2 ± 0.72 | 9.3 ± 0.67 |

## 4.4. Evaluation

We evaluated model performance using top-1 and top-5 image retrieval accuracy. Specifically, during testing, we computed the similarity of each test brain embedding and the image embeddings of all 200 unseen images in the test set, then ranked them accordingly. The top-$k$ accuracy measures whether the correct image appears among the $k$ most similar image embeddings to the given brain embedding.

## 5. Results

We conduct an extensive empirical study comparing the performance of human-aligned and unaligned image encoders in visual decoding from brain activity. Specifically, we assess their impact across various image encoder architectures and alignment methods (Section 5.1), as well as various EEG encoders (Section 5.2). We also assess whether human-aligned models generalize better to unseen subjects (Section 5.3) and examine their effectiveness in other neuroimaging modalities, such as MEG (Section 5.4). Finally, we perform an interpretability analysis (Section 5.5) to highlight differences between EEG models trained with human-aligned and original image embeddings. Overall, our results highlight the consistent improvement of using human-aligned models for visual decoding from the brain.

## 5.1. Image Encoders and Alignment Methods

We start by evaluating how different image encoders and alignment methods affect visual decoding performance from EEG activity. In particular, we consider the *Dreamsim* (Fu et al., 2023; Sundaram et al., 2024), *gLocal* (Muttenthaler et al., 2023a) and *Harmonization* (Fel et al., 2022) alignment

methods. For consistency, we employ the *NICE* EEG encoder (Song et al., 2024) across all evaluations. We present the main results in Figure 3.

EEG encoders trained with human-aligned embeddings using *Dreamsim* consistently and significantly outperform those trained with unaligned embeddings, regardless of the type of image encoder used (Fig. 3a). Notably, these human-aligned models achieve up to 21% higher top-1 and 29% higher top-5 accuracy in 200-way retrieval tasks. A paired T-test further confirms the significance of these improvements ($p < 0.001$ for all except *DINO* and *Ensemble*, where $p < 0.01$ and $p < 0.05$, respectively, for the top-1 performance).

Both *gLocal* and *Harmonization* perform comparably to original unaligned models and sometimes even surpass them (Fig. 3b, Fig. 3c). However, they still underperform relative to *Dreamsim*. We attribute this to differences in the datasets used to align these models with human perception. *Dreamsim* is trained on image similarity judgment datasets that capture mid-level similarities, which are more likely to be reflected in brain data collected via RSVP. In contrast, *gLocal* and *Harmonization* emphasize similarities that require more attention and longer viewing times. We further discuss these differences in Section 6.

## 5.2. EEG Architecture

We now assess the impact of different EEG architectures on the overall performance of the method. We evaluate the *EEGNetV4* (Lawhern et al., 2018), *EEGConformer* (Song et al., 2022), *NICE* (Song et al., 2024), and *ATM-S* (Li et al., 2024) as EEG encoders in our evaluation. As before, base-

*Table 3.* **200-way top-1 image retrieval performance from MEG signals.** We encode MEG using the *NICE* encoder and consider different image encoders during the training of the models. The results show that using human-aligned (HA) image encoders consistently improves the performance of the overall system, regardless of the specific image encoder employed. Best viewed with zoom.

| | Ensemble | | SynCLR | | CLIP | | OpenCLIP | | DINO | | DINOv2 | |
|---|---|---|---|---|---|---|---|---|---|---|---|---|
| | HA | Base | HA | Base | HA | Base | HA | Base | HA | Base | HA | Base |
| S1 | $13.4 \pm 1.82$ | $12.1 \pm 1.88$ | $15.5 \pm 1.14$ | $13.0 \pm 1.30$ | $12.6 \pm 0.86$ | $7.1 \pm 0.49$ | $11.4 \pm 1.11$ | $8.7 \pm 0.68$ | $10.7 \pm 1.03$ | $11.8 \pm 1.75$ | $11.7 \pm 2.02$ | $7.5 \pm 1.10$ |
| S2 | $36.7 \pm 2.56$ | $29.6 \pm 0.97$ | $36.8 \pm 2.54$ | $24.5 \pm 1.52$ | $28.3 \pm 2.75$ | $13.8 \pm 1.81$ | $29.6 \pm 2.11$ | $20.6 \pm 2.27$ | $33.6 \pm 1.56$ | $24.75 \pm 2.61$ | $30.6 \pm 1.28$ | $12.3 \pm 1.17$ |
| S3 | $22.7 \pm 1.08$ | $20.6 \pm 1.02$ | $25.6 \pm 2.41$ | $16.9 \pm 1.62$ | $22.2 \pm 1.63$ | $13.4 \pm 1.32$ | $21.4 \pm 0.97$ | $15.4 \pm 1.39$ | $21.1 \pm 1.77$ | $19.12 \pm 1.02$ | $17.5 \pm 0.89$ | $7.6 \pm 0.97$ |
| S4 | $14.8 \pm 1.96$ | $12.2 \pm 1.72$ | $14.7 \pm 2.94$ | $12.7 \pm 2.45$ | $14.3 \pm 1.33$ | $8.1 \pm 0.73$ | $11.1 \pm 1.68$ | $10.9 \pm 2.08$ | $10.2 \pm 1.17$ | $10.2 \pm 1.48$ | $10.8 \pm 1.75$ | $5.8 \pm 1.78$ |
| Avg. | $\mathbf{21.9 \pm 5.34}$ | $18.6 \pm 4.16$ | $\mathbf{23.2 \pm 5.18}$ | $16.8 \pm 2.75$ | $\mathbf{19.3 \pm 3.64}$ | $10.6 \pm 1.75$ | $\mathbf{18.4 \pm 4.44}$ | $13.9 \pm 2.66$ | $\mathbf{18.9 \pm 5.51}$ | $16.5 \pm 3.37$ | $\mathbf{17.7 \pm 4.56}$ | $8.3 \pm 1.40$ |

line embeddings are compared with human-aligned embeddings from multiple image encoders. We consider *Dreamsim* as the alignment method for this evaluation. The results are summarized in Table 1.

Once again, we observe a consistent and significant improvement when we consider the human-aligned version of the image embeddings during training. In particular, human-alignment allows us to achieve a 62% top-1 accuracy, a significant improvement over the 46% accuracy achieved by non-aligned versions. As expected from prior literature, certain architectures (*NICE*, *ATM-S*) perform better overall than others (*EEGNetV4*, *EEGConformer*). Nonetheless, encoders trained with human-aligned embeddings consistently outperformed those trained with unaligned embeddings, regardless of the underlying EEG architecture.

### 5.3. Transferring to New Subjects

Generalization of trained models to unseen participants is a critical yet challenging task in brain data studies, as the subjective nature of brain signals often leads to poor performance on new participants' data. In this experiment, we compared the naive generalization capability of the *NICE* EEG encoder trained with human-aligned embeddings to its unaligned counterpart. Specifically, the *NICE* encoder was trained using data from 9 participants and tested on the held-out participant's test data. The results for each test participant and image encoder are presented in Table 2.

The results highlight the challenge of transferring brain signals to unseen participants. A paired T-test showed human-aligned models significantly outperformed original models for *CLIP* ($p < 0.001$) and *OpenCLIP* ($p < 0.05$), while no significant differences were found for other architectures. Though improving transferability is beyond the scope of this work, we plan to explore strategies for enhancing generalization in future research.

### 5.4. Validation on MEG Signals

To evaluate whether the performance improvement of using human-aligned image encoders extends to other neuroimaging modalities, we replicate the experimental procedure of Section 5.1, this time using MEG data. Specifically, we utilized the MEG dataset from Hebart et al. (2023), as de-

scribed in Section 4.1, and employed a similar architecture to the *NICE* encoder. We present the top-1 retrieval performances, discriminated by subject, in Table 3, and top-5 performance in Appendix D.

Results consistently show improved retrieval performance across all participants using human-aligned embeddings with *Dreamsim* over original image encoders. While statistical tests were not conducted due to the limited number of participants, the findings highlight the potential of human-aligned image encoders to enhance visual decoding across brain activity modalities.

Finally, we also evaluated the use of human-aligned models for image retrieval from fMRI signals (Allen et al., 2022), with results presented in Appendix E.

### 5.5. Biological Interpretation

To investigate whether the improved performance with perceptually aligned embeddings is linked to attributes of visual processing in the brain, we analyzed the gradients of the EEG encoder from spatial, temporal, and spectral perspectives. Specifically, we applied gradient-weighted class activation mapping (Grad-CAM) (Selvaraju et al., 2017) to identify the regions in the signals that contributed most to the model's predictions. Figure 4a presents the histogram of gradient values exceeding the $99^{th}$ percentile over time for a representative participant. Notably, EEG encoders trained with human-aligned image embeddings exhibit a higher peak at earlier time points ($0 - 200$ ms relative to stimulus onset). This period overlaps with the image stimulus presentation time and is associated with low-level visual processing, such as orientation and color (Lamme & Roelfsema, 2000; Hillyard & Anllo-Vento, 1998). Figure 5 further supports this, demonstrating that human-aligned models, compared to original ones, more accurately retrieve attributes like texture, pattern, color, and shape, which are consistently shared among the top-5 retrieved images. Additional examples are provided in Appendix F.

EEG signals are typically decomposed into five frequency bands linked to cognitive processes: Delta ($[0 - 4]$ Hz), Theta ($[4 - 8]$ Hz), Alpha ($[8 - 12]$ Hz), Beta ($[12 - 30]$ Hz), and Gamma ($> 30$ Hz)). To assess each band's contribution, we computed the power spectral density of the

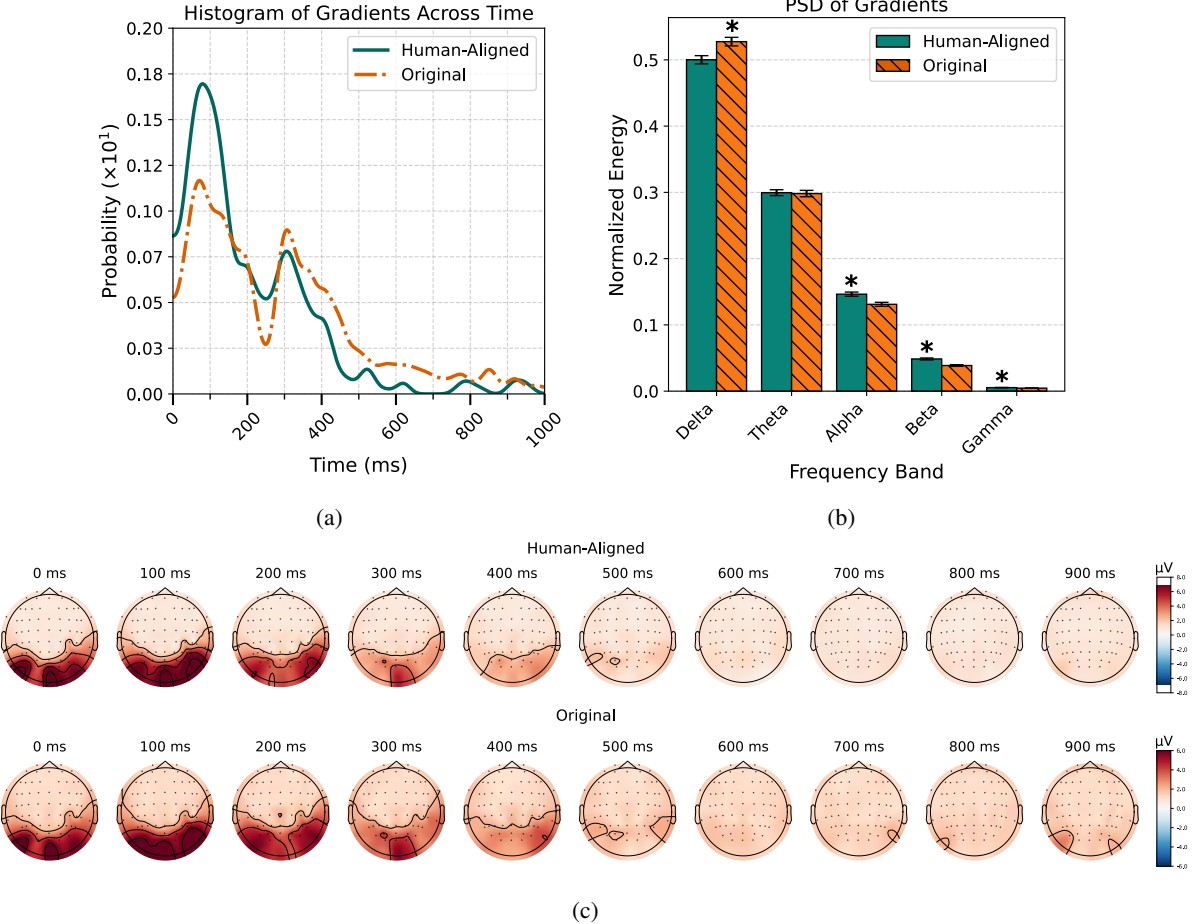

*Figure 4.* **Analyzing gradient maps for *NICE* EEG Encoder**. We computed the gradients for trained EEG encoders with human-aligned and original unaligned image embeddings and analyzed them from (a) temporal, (b) spectral, and (c) spatial perspectives. The results are presented for a representative participant (*S4*) and image backbone (*SynCLR*).

gradients and normalized them by total power. Fig. 4b compares the energy of each band across human-aligned and original models, with statistical significance assessed via an unpaired T-test ($p < 0.005$ for Delta, Alpha, and Beta; $p < 0.05$ for Gamma). Models trained with original embeddings focused more on Delta, while human-aligned models emphasized Alpha, Beta, and Gamma, aligning with prior research linking these bands to visual processing (Bastos et al., 2015; Michalareas et al., 2016).

We also examined which EEG electrodes contribute more to the predictions of the model. Both EEG encoders, whether trained with human-aligned or original image embeddings, relied on information from electrodes positioned over the occipital and parietal brain regions, which are known to be involved in visual processing (Grill-Spector & Malach, 2004; Wandell et al., 2007) (Figure 4c). From this perspective, no significant differences were observed between the models. More results and details are provided in Appendix G.

# 6. Discussion

In this paper, we investigate the use of human-aligned image models for decoding visual information from brain signals. We demonstrate that brain encoders trained with human-aligned image embeddings consistently outperform those trained with unaligned embeddings in image retrieval tasks. This finding is validated through extensive evaluations across various image and brain encoders, alignment methods, and brain modalities. Finally, we provide some insights on spatial, temporal, and spectral differences between the models by analyzing the gradients of the brain encoder.

**Perceptual Alignment and Brain Decoding.** Human similarity judgments stem from brain processes and reflect the similarities between brain signals elicited by image stimuli. To evaluate the effect of using alignment, in Section 5.5 we analyze gradient maps produced by the brain encoder across three domains: spatial, temporal, and spectral. The spatial analysis showed that both human-aligned and non-human-

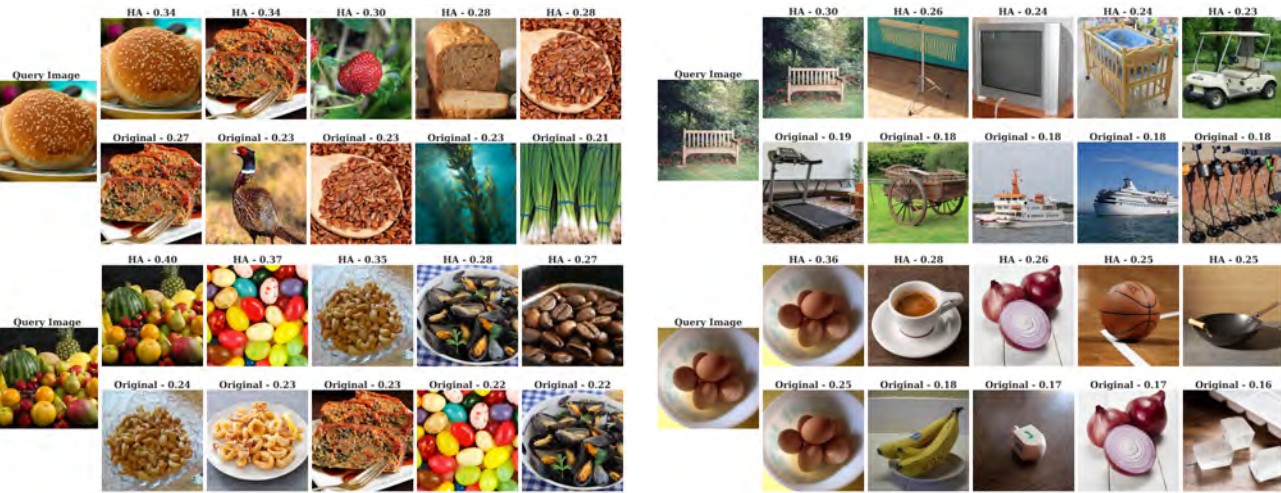

*Figure 5.* **Top-5 retrieved images of unseen classes from EEG signals using human-aligned (top) and base (bottom) models.** We use *Dreamsim* for the aligned model with *CLIP* backbone for both. The cosine similarity between EEG and image embeddings is shown above the images. Images retrieved from human-aligned models emphasize attributes like texture, pattern, color, and shape.

aligned models focus on similar brain regions, particularly the visual cortex. However, differences emerged in the temporal and spectral domains, with the human-aligned model placing greater emphasis on early stimulus responses and frequency components in Alpha, Beta, and Gamma bands.

We attribute this behavior to the experimental design of the image-similarity dataset. Brain signals, elicited by rapid visual stimuli (100 ms for EEG and 500 ms for MEG), primarily capture early impressions of images, such as shape, orientation, and color. These features are better represented in image-similarity datasets like *NIGHTS*, which emphasize mid-level similarities, compared to those focusing on fine-grained categorical differences. As a result, image encoders fine-tuned with such datasets are more aligned to differences between images that are also reflected in the recorded brain signals. We visualized some of the retrieved images from EEG signals in Figure 5, showing that the retrieved images for the human-aligned models indeed share some attributes such as color, shape, and orientation.

**Why *Dreamsim* outperforms other alignment methods.** We also evaluated embeddings generated by human-aligned image encoders trained with earlier methods. While these models performed similarly to those using unaligned embeddings, they did not achieve the substantial improvements observed with *Dreamsim* models. This difference likely arises from variations in datasets and training methodologies. For example, *gLocal* relies on *THINGS* odd-one-out triplets (Hebart et al., 2019), which emphasize high-level category differences, whereas *Dreamsim* uses the *NIGHTS* (Fu et al., 2023) similarity judgment dataset, capturing mid-level attributes such as object appearance, viewing angles, camera poses, and layout. Unlike earlier datasets focused on

low-level perturbations (e.g., blurring, noise) or high-level categorical distinctions, *NIGHTS* prioritizes features that are also relevant to rapid visual processing, aligning more closely with brain-computer interaction tasks in this study. Finally, *Harmonization* leverages the *ClickMe* (Linsley et al., 2019) and *Clicktionary* (Linsley et al., 2017) datasets, which use human-annotated saliency maps to train image encoders aligned with human perception. However, tasks in *ClickMe* and *Clicktionary* require careful inspection and attention to images—conditions that do not reflect the rapid visual stimulus used in brain data collection experiments. As a result, *Harmonization* models are less likely to capture the same features as the brain datasets used in this study.

**Future Work.** Our work opens several new possibilities for future work. We mainly looked at this problem from the perspective of image retrieval to provide a more direct and easier-to-evaluate scenario that allows us to isolate the effect of human alignment better. Another important downstream task is reconstructing viewed images from brain activity using image generative models like Stable Diffusion (Rombach et al., 2022). This venue of exploration could be achieved by fine-tuning the image generation process on a human-aligned conditioning representation space. Moreover, methods for quantifying the alignment (Sucholutsky et al., 2023) can provide significant insight and could also be explored. Finally, to generalize our findings, future work should validate the results on datasets featuring brain responses to images from other databases. We plan to extend this study to fMRI, which offers higher spatial resolution and longer stimulus times, to test the consistency of our findings.

**Conclusion.** In this work, we proposed that human-aligned models consistently and significantly improve vi-

sual decoding from the brain. Our extensive empirical study highlighted the pivotal role of image representations and experimental design in visual decoding from brain data. We hope our study introduces a new perspective on learning brain-to-image mappings and lays the groundwork for future advances in visual decoding from brain activity.

## Acknowledgments

This work was supported by the Knut and Alice Wallenberg Foundation, the Swedish Research Council, European Research Council (ERC) D2Smell 101118977 and ERC BIRD 88480 grants. The computations and data handling were enabled mainly by the Berzelius resource provided by the Knut and Alice Wallenberg Foundation at the National Supercomputer Centre and partly by the National Academic Infrastructure for Supercomputing in Sweden (NAISS), partially funded by the Swedish Research Council through grant agreement no. 2022-06725.

We would also like to thank Mohammed Al-Jaff for his valuable insights and constructive discussions.

## Impact Statement

Decoding images from brain signals has seen significant progress in recent years. However, this technology remains limited to research environments and is not yet suitable for applications outside controlled laboratory settings. These models remain in the research phase, and no consequential decisions should be based on their outputs. The technology is not yet reliable enough for any form of critical application, and its results should be interpreted with caution. An application of this research could be in brain-computer interaction, particularly for assistive technologies aiding individuals with communication disabilities. While these advancements could enhance accessibility, current models remain untested on patient data. Further research is needed to validate their use in medical settings.

Given that human data is used in this research, ethical approval is a necessary prerequisite for data collection. In our work, we exclusively utilized publicly available datasets that were ethically reviewed and approved by official organizations. Furthermore, all data used in this study has been anonymized to ensure the privacy and confidentiality of participants, aligning with best practices in ethical research.

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

## A. Datasets Details

**EEG Data** We used EEG data from Gifford et al. (2022) and preprocessed it in the same way as Song et al. (2024); Li et al. (2024). Particularly, first, a multivariate noise normalization (Guggenmos et al., 2018) was applied to the data from each session before epoching the signals. Then, signals from 63 EEG electrodes were filtered to the range [0.1–100] Hz and downsampled to 250 Hz. EEG data was epoched from 0 to 1000 ms relative to the stimulus onset, with baseline correction applied using the 200 ms pre-stimulus interval.

**MEG Data** For the MEG experiments, we used the preprocessed MEG data from Hebart et al. (2023). The dataset includes recordings from four participants with 271 channels. Signals were band-passed between 0.1 and 40 Hz and baseline corrected by subtracting the mean and dividing by the standard deviation of the 100 ms period before stimulus onset. While the original data was epoched from -100 to 1300 ms relative to stimulus onset, we used a subset from 0 to 1000 ms. The data was initially sampled at 1200 Hz but was downsampled to 200 Hz during preprocessing.

## B. Implementation Details

**Aligned Image Encoders** We obtained human-aligned image encoders directly from the *Dreamsim*, *gLocal*, and *Harmonization* repositories provided by the authors, ensuring they function exactly as reported in their respective papers without any retraining. The code for generating embeddings for images in the *THINGS-EEG2* and *THINGS-MEG2* datasets is available in the project repository.

**Unaligned Image Encoders** For the original unaligned encoders, we used publicly available pretrained models using *Hugging Face transformers* (Table 4) or *timm* (Table 5) libraries.

| Model Name | Hugging Face Repository Path |
|---|---|
| CLIP_ViT-B32 | openai/clip-vit-base-patch32 |
| CLIP_ViT-B16 | openai/clip-vit-base-patch16 |
| CLIP_ViT-L14 | openai/clip-vit-large-patch14 |
| DINO_ViT-B16 | facebook/dino-vitb16 |
| DINO_ViT-B8 | facebook/dino-vitb8 |
| DINOv2_ViT-B14 | facebook/dinov2-base |
| DINOv2_ViT-L14 | facebook/dinov2-large |
| OpenCLIP_ViT-L14_laion2b | hf-hub:laion/CLIP-ViT-L-14-laion2B-s32B-b82K |
| OpenCLIP_ViT-L14_laion400m | hf-hub:timm/vit_large_patch14_clip_224.laion400m_e32 |
| OpenCLIP_ViT-B32_laion400m | hf-hub:timm/vit_base_patch32_clip_224.laion400m_e31 |
| OpenCLIP_RN50 | hf-hub:timm/resnet50_clip.openai |

*Table 4.* Model names and their corresponding Hugging Face repository paths.

| Model Type | Timm Model Name |
|---|---|
| ViT-B16 | vit_base_patch16_224 |
| ResNet50 | resnet50 |
| VGG16 | vgg16 |
| ConvNEXT | convnext_tiny |
| LeViT | levit_conv_128 |

*Table 5.* Mapping of model types to their corresponding `timm` model names.

**Training** We performed a grid search on a sample subject to determine the optimal learning rate, batch size, and temperature parameter, selecting the best values based on minimum validation loss. For per-participant EEG experiments, *NICE* encoders were trained for up to 50 epochs with a batch size of 128, a learning rate of 0.0002, and a temperature of 0.04. The same hyperparameters were used for *ATM-S*, except for the number of epochs, which was set to 80. *EEGNet* and

*EEGConformer* were both trained for 200 epochs. *EEGConformer* used a learning rate of 0.0002, a batch size of 128, and a temperature of 0.07, while *EEGNet* used 0.01, 512, and 0.1, respectively. For cross-participant training, *NICE* was trained for up to 150 epochs with a batch size of 512 and a learning rate of 0.0001. For MEG experiments, we used a learning rate of 0.00005, a batch size of 256, and a temperature of 0.1, training for up to 50 epochs. Training was halted in all models if validation loss did not improve for 25 consecutive epochs.

## C. Cross-Subject EEG Top-5 Retrieval Performances

Table 6 presents the top-5 performance of the experiment presented in Section 5.3.

*Table 6.* Cross-subject 200-way top-5 image retrieval performance of *NICE* encoder on EEG signals using human-aligned or original image embeddings.

|  | Ensemble | | SynCLR | | CLIP | | OpenCLIP | | DINO | | DINOv2 | |
|---|---|---|---|---|---|---|---|---|---|---|---|---|
|  | HA | Base | HA | Base | HA | Base | HA | Base | HA | Base | HA | Base |
| S1 | $33.6 \pm 2.6$ | $31.5 \pm 1.0$ | $32.6 \pm 2.7$ | $27.0 \pm 1.7$ | $33.7 \pm 2.8$ | $25.0 \pm 1.9$ | $33.9 \pm 3.2$ | $25.4 \pm 0.9$ | $32.3 \pm 2.9$ | $28.2 \pm 1.3$ | $29.3 \pm 0.9$ | $29.1 \pm 1.6$ |
| S2 | $38.1 \pm 1.6$ | $38.4 \pm 1.4$ | $34.1 \pm 3.4$ | $32.8 \pm 0.7$ | $32.0 \pm 0.8$ | $25.2 \pm 2.0$ | $30.1 \pm 3.4$ | $27.8 \pm 2.3$ | $35.3 \pm 1.3$ | $33.3 \pm 1.7$ | $34.2 \pm 2.4$ | $27.6 \pm 1.6$ |
| S3 | $28.3 \pm 1.4$ | $31.9 \pm 4.7$ | $33.5 \pm 1.9$ | $31.9 \pm 3.2$ | $30.7 \pm 3.3$ | $23.9 \pm 2.3$ | $33.2 \pm 4.4$ | $24.6 \pm 0.9$ | $30.8 \pm 1.4$ | $29.9 \pm 2.4$ | $26.6 \pm 1.6$ | $25.0 \pm 2.5$ |
| S4 | $30.1 \pm 3.9$ | $33.6 \pm 1.9$ | $29.8 \pm 2.4$ | $30.1 \pm 3.6$ | $28.6 \pm 2.0$ | $24.8 \pm 1.5$ | $26.7 \pm 1.6$ | $24.2 \pm 1.5$ | $27.9 \pm 1.7$ | $27.3 \pm 2.2$ | $27.3 \pm 1.8$ | $27.6 \pm 1.5$ |
| S5 | $23.5 \pm 0.8$ | $23.0 \pm 1.1$ | $21.9 \pm 2.0$ | $21.6 \pm 1.4$ | $22.1 \pm 2.1$ | $16.5 \pm 1.3$ | $23.3 \pm 3.7$ | $20.5 \pm 3.6$ | $21.6 \pm 1.9$ | $22.2 \pm 1.8$ | $19.3 \pm 3.4$ | $16.9 \pm 1.4$ |
| S6 | $46.9 \pm 1.3$ | $41.1 \pm 1.4$ | $41.2 \pm 2.6$ | $36.7 \pm 3.7$ | $38.2 \pm 2.8$ | $32.9 \pm 1.4$ | $36.5 \pm 3.4$ | $32.3 \pm 1.6$ | $36.2 \pm 1.9$ | $36.8 \pm 1.3$ | $34.7 \pm 2.8$ | $26.3 \pm 1.7$ |
| S7 | $35.2 \pm 2.6$ | $32.6 \pm 1.1$ | $29.8 \pm 1.6$ | $29.1 \pm 2.0$ | $28.8 \pm 3.2$ | $24.5 \pm 1.6$ | $27.3 \pm 0.9$ | $27.4 \pm 2.4$ | $32.0 \pm 3.3$ | $31.2 \pm 2.5$ | $25.9 \pm 3.1$ | $20.0 \pm 1.8$ |
| S8 | $35.6 \pm 2.9$ | $35.3 \pm 3.4$ | $35.3 \pm 2.5$ | $34.5 \pm 1.3$ | $31.6 \pm 3.2$ | $28.2 \pm 1.3$ | $35.9 \pm 2.3$ | $31.4 \pm 0.9$ | $35.1 \pm 2.1$ | $31.5 \pm 3.2$ | $33.6 \pm 2.0$ | $26.3 \pm 1.2$ |
| S9 | $33.8 \pm 2.9$ | $33.6 \pm 5.8$ | $29.5 \pm 2.0$ | $30.2 \pm 2.0$ | $27.1 \pm 3.5$ | $24.1 \pm 2.6$ | $26.4 \pm 1.2$ | $26.1 \pm 0.9$ | $32.4 \pm 1.8$ | $31.2 \pm 1.1$ | $28.1 \pm 2.1$ | $25.3 \pm 2.3$ |
| S10 | $40.8 \pm 4.3$ | $40.2 \pm 1.9$ | $33.4 \pm 2.6$ | $32.7 \pm 1.3$ | $33.1 \pm 1.5$ | $28.4 \pm 2.7$ | $36.2 \pm 1.6$ | $32.2 \pm 2.3$ | $32.1 \pm 3.4$ | $33.9 \pm 3.2$ | $28.2 \pm 2.3$ | $31.9 \pm 2.5$ |
| Avg. | $34.6 \pm 2.4$ | $34.1 \pm 2.4$ | $32.1 \pm 2.4$ | $30.7 \pm 2.1$ | $\mathbf{30.6 \pm 2.5}$ | $25.3 \pm 1.9$ | $\mathbf{30.9 \pm 2.6}$ | $27.2 \pm 1.7$ | $31.6 \pm 2.2$ | $30.5 \pm 2.1$ | $28.7 \pm 2.2$ | $25.6 \pm 1.8$ |

## D. MEG Top-5 Retrieval Performances

Table 7 presents the top-5 performance of the experiment presented in Section 5.4.

*Table 7.* 200-way zero-shot top-5 performance of the *NICE* encoder on MEG data using human-aligned or original image embeddings.

|  | Ensemble | | SynCLR | | CLIP | | OpenCLIP | | DINO | | DINOV2 | |
|---|---|---|---|---|---|---|---|---|---|---|---|---|
|  | HA | Base | HA | Base | HA | Base | HA | Base | HA | Base | HA | Base |
| S1 | $40.1 \pm 3.7$ | $40.1 \pm 1.8$ | $39.6 \pm 1.2$ | $35.6 \pm 1.8$ | $35.7 \pm 3.1$ | $26.6 \pm 0.9$ | $35.9 \pm 2.4$ | $29.0 \pm 1.1$ | $35.7 \pm 1.1$ | $34.1 \pm 2.3$ | $33.4 \pm 1.4$ | $21.2 \pm 1.9$ |
| S2 | $72.7 \pm 1.4$ | $62.9 \pm 1.3$ | $75.2 \pm 1.6$ | $62.1 \pm 2.0$ | $62.5 \pm 2.3$ | $44.7 \pm 1.4$ | $64.5 \pm 1.2$ | $53.1 \pm 1.5$ | $66.5 \pm 1.9$ | $58.1 \pm 1.9$ | $60.7 \pm 3.6$ | $34.1 \pm 1.5$ |
| S3 | $58.6 \pm 2.0$ | $49.6 \pm 1.8$ | $55.0 \pm 2.5$ | $47.7 \pm 2.3$ | $48.8 \pm 1.9$ | $36.0 \pm 1.9$ | $53.3 \pm 3.0$ | $41.8 \pm 2.1$ | $52.1 \pm 1.9$ | $42.9 \pm 2.8$ | $46.8 \pm 4.0$ | $23.1 \pm 2.5$ |
| S4 | $36.4 \pm 2.1$ | $35.1 \pm 1.4$ | $38.6 \pm 0.8$ | $31.9 \pm 3.7$ | $36.7 \pm 1.8$ | $26.4 \pm 2.1$ | $32.9 \pm 1.3$ | $33.5 \pm 3.2$ | $37.3 \pm 4.0$ | $30.6 \pm 2.6$ | $32.4 \pm 1.0$ | $19.4 \pm 1.1$ |
| Avg. | $\mathbf{51.9 \pm 2.3}$ | $46.9 \pm 1.6$ | $\mathbf{52.1 \pm 1.5}$ | $44.3 \pm 2.5$ | $\mathbf{45.9 \pm 2.3}$ | $33.4 \pm 1.6$ | $\mathbf{46.6 \pm 2.0}$ | $39.3 \pm 2.0$ | $\mathbf{47.9 \pm 2.2}$ | $41.4 \pm 2.4$ | $\mathbf{43.3 \pm 2.5}$ | $24.4 \pm 1.8$ |

## E. Evaluation on fMRI Data

To further validate the benefits of using human-aligned models, we extended our experiments to the NSD dataset (Allen et al., 2022). This dataset contains 7-Tesla fMRI responses of human participants to natural scenes from the MS-COCO image database (Lin et al., 2014). We followed the same preprocessing and data splitting steps as Takagi & Nishimoto (2023); Scotti et al. (2023; 2024) and refer the reader to those works for further details. We trained individual-subject models for the four participants who completed all scanning sessions (participants 1, 2, 5, and 7), using a test set based on the 1,000 images that were presented to all participants. This resulted in a dataset comprising 24,980 training samples and 2,770 test samples. For the test set, we averaged the brain responses across the three repetitions of each image, reducing the test set to 982 unique samples, while the training set remained unaveraged.

We trained the MLP fMRI encoder with residual connections proposed by Scotti et al. (2023) for 50 epochs with a learning rate of 0.0001 and a batch size of 128. Top-1 and top-5 1000-way retrieval accuracies are presented in Tables 8 and 9, respectively.

*Table 8.* 1000-way zero-shot top-1 performance of the *MLP* encoder on fMRI data using human-aligned or original image embeddings.

|  | Ensemble | | SynCLR | | CLIP | | OpenCLIP | | DINO | | DINOv2 | |
|---|---|---|---|---|---|---|---|---|---|---|---|---|
|  | HA | Base | HA | Base | HA | Base | HA | Base | HA | Base | HA | Base |
| S1 | $60.4 \pm 0.9$ | $52.9 \pm 1.2$ | $63.5 \pm 1.1$ | $52.0 \pm 0.7$ | $49.2 \pm 1.4$ | $27.7 \pm 1.7$ | $53.6 \pm 0.6$ | $40.2 \pm 0.8$ | $53.7 \pm 1.3$ | $49.9 \pm 1.4$ | $50.0 \pm 1.5$ | $24.8 \pm 0.8$ |
| S2 | $52.9 \pm 0.8$ | $44.3 \pm 0.9$ | $57.5 \pm 0.9$ | $43.9 \pm 1.0$ | $43.3 \pm 1.0$ | $21.6 \pm 0.8$ | $48.6 \pm 0.9$ | $31.5 \pm 0.7$ | $47.9 \pm 0.9$ | $43.0 \pm 0.5$ | $42.6 \pm 1.1$ | $20.2 \pm 0.5$ |
| S5 | $62.2 \pm 1.2$ | $56.5 \pm 1.3$ | $66.2 \pm 0.2$ | $57.8 \pm 0.7$ | $53.7 \pm 0.6$ | $27.7 \pm 0.9$ | $57.9 \pm 1.0$ | $43.2 \pm 0.6$ | $55.5 \pm 1.2$ | $53.7 \pm 1.4$ | $52.2 \pm 0.7$ | $27.2 \pm 1.3$ |
| S7 | $43.1 \pm 1.3$ | $38.8 \pm 1.2$ | $45.6 \pm 1.0$ | $38.7 \pm 1.3$ | $33.8 \pm 0.6$ | $18.0 \pm 0.6$ | $37.8 \pm 1.7$ | $27.0 \pm 1.2$ | $39.3 \pm 0.9$ | $37.0 \pm 1.1$ | $36.5 \pm 0.5$ | $17.9 \pm 0.6$ |
| Avg. | $\mathbf{54.6 \pm 4.3}$ | $48.1 \pm 4.0$ | $\mathbf{58.2 \pm 4.6}$ | $48.2 \pm 4.3$ | $\mathbf{45.0 \pm 4.3}$ | $23.8 \pm 2.4$ | $\mathbf{49.5 \pm 4.3}$ | $35.4 \pm 3.8$ | $\mathbf{49.1 \pm 3.7}$ | $45.9 \pm 3.7$ | $\mathbf{45.3 \pm 3.6}$ | $22.5 \pm 2.1$ |

*Table 9.* 1000-way zero-shot top-5 performance of the *MLP* encoder on fMRI data using human-aligned or original image embeddings.

|  | Ensemble | | SynCLR | | CLIP | | OpenCLIP | | DINO | | DINOv2 | |
|---|---|---|---|---|---|---|---|---|---|---|---|---|
|  | HA | Base | HA | Base | HA | Base | HA | Base | HA | Base | HA | Base |
| S1 | $87.8 \pm 0.4$ | $83.3 \pm 0.6$ | $89.4 \pm 0.9$ | $81.8 \pm 0.9$ | $79.9 \pm 0.6$ | $59.4 \pm 0.9$ | $83.2 \pm 0.3$ | $72.4 \pm 0.7$ | $83.6 \pm 1.0$ | $80.6 \pm 0.5$ | $79.4 \pm 0.6$ | $51.7 \pm 1.5$ |
| S2 | $82.5 \pm 1.1$ | $77.84 \pm 1.0$ | $85.7 \pm 0.7$ | $76.0 \pm 0.5$ | $76.0 \pm 1.3$ | $50.8 \pm 0.3$ | $78.0 \pm 0.5$ | $64.7 \pm 0.8$ | $79.1 \pm 0.8$ | $73.6 \pm 0.3$ | $74.7 \pm 0.6$ | $45.0 \pm 1.5$ |
| S5 | $89.9 \pm 0.8$ | $86.5 \pm 1.1$ | $91.9 \pm 0.3$ | $85.3 \pm 0.7$ | $84.9 \pm 0.3$ | $61.5 \pm 0.9$ | $87.2 \pm 0.6$ | $76.8 \pm 0.8$ | $86.7 \pm 0.7$ | $84.6 \pm 0.5$ | $82.0 \pm 0.2$ | $56.9 \pm 1.0$ |
| S7 | $73.5 \pm 1.0$ | $70.8 \pm 1.6$ | $75.7 \pm 0.9$ | $71.5 \pm 0.8$ | $65.4 \pm 0.4$ | $43.7 \pm 1.8$ | $68.4 \pm 0.8$ | $57.4 \pm 0.5$ | $69.6 \pm 0.4$ | $68.2 \pm 0.9$ | $65.9 \pm 0.5$ | $43.0 \pm 0.6$ |
| Avg. | $\mathbf{83.4 \pm 3.7}$ | $79.6 \pm 3.4$ | $\mathbf{86.7 \pm 3.6}$ | $78.7 \pm 3.1$ | $\mathbf{76.6 \pm 4.2}$ | $53.8 \pm 4.1$ | $\mathbf{79.2 \pm 4.1}$ | $67.8 \pm 4.3$ | $\mathbf{79.7 \pm 3.7}$ | $76.8 \pm 3.6$ | $\mathbf{75.5 \pm 3.5}$ | $49.1 \pm 3.2$ |

## F. More Examples of Retrieved Images

In addition to the original *Things EEG2* test set of 200 samples, we explored retrieved images from a larger dataset. Specifically, using EEG data from the test set, we computed the distances between EEG embeddings and all image embeddings from both the training and test sets, expanding the search space to 16,740 images with a chance performance of 0.006%. This broader search provides deeper insights into the neighboring images of an EEG sample and highlights differences between the representation spaces learned with human-aligned and unaligned image embeddings.

Figure 6 presents examples of the top-5 retrieved images from this larger dataset. The results suggest that human-aligned (*Dreamsim*) models are more sensitive to both color and pattern. As seen in Figures 6e, 6f, 6g, and 6i, while both the top (human-aligned) and bottom (original) rows contain images with similar patterns to the ground truth (query) image, those retrieved by human-aligned models also share similar colors with the ground truth. Additionally, while the original models sometimes better estimate object categories, human-aligned models appear to be more effective in retrieving the correct orientation and overall shape of the object (see Figures 6a, 6b, 6c, 6d, 6h, 6j). Table 10 show the retrieval performance when using the large image set for a representative subject. The results show that for human-aligned models the top-1 performance can achieve up to 12.5% which is significantly higher than the chance performance (0.006%).

We also visualized some of the retrieved images using human-aligned models with *gLocal* as the alignment method, as shown in Figure 7. Interestingly, images retrieved from *gLocal* model embeddings exhibit greater consistency in object category compared to both the original unaligned versions and *Dreamsim* models. However, unlike *Dreamsim*, images retrieved from *gLocal* embeddings do not necessarily match the ground truth image in terms of orientation and color (see Figures 7b, 7c, 7d, and 7f).

This highlights an important point: each human-aligned model prioritizes attributes emphasized in the dataset used for alignment with human perception, which directly influences their learned representations (as discussed in Section 5.5). Additionally, the way participants experience image stimuli during visual brain data collection affects their perception of the images. Currently available datasets have been collected using the RSVP paradigm, which primarily captures low-level and mid-level features. As a result, they align more closely with *Dreamsim*'s approach. Future work should explore alternative experimental designs for brain data collection and investigate their relationship with different human-alignment methods.

*Table 10.* 200-way zero-shot performance of the *NICE* encoder on EEG data of a representative subject using a larger image database for retrieval. numbers show top-1 accuracies and top-5 in paranthesis.

|  | Ensemble | | SynCLR | | CLIP | | OpenCLIP | | DINO | | DINOv2 | |
|---|---|---|---|---|---|---|---|---|---|---|---|---|
|  | HA | Base | HA | Base | HA | Base | HA | Base | HA | Base | HA | Base |
| S10 | 9.5 (22.0) | 3.0 (12.0) | 12.5 (28.0) | 3.0 (12.0) | 4.5 (14.5) | 3.5 (8.5) | 6.0 (19.5) | 4.5 (11.5) | 7.5 (24.0) | 3.5 (12.5) | 3.0(10.0) | 0.5(2.0) |

# G. Grad-CAM Analysis Per Participant

## G.1. Temporal Analysis

In Section 5.5, we plotted a histogram of large gradient values over time for a representative participant and image encoder. We extracted gradients from the projection layer of the *NICE* model's convolutional encoder (`encoder.0.projection.0`) for models trained with five different seeds on one participant's data. Only positive gradient values were kept, highlighting features that positively contributed to predictions. These gradients were then normalized between 0 and 1 and interpolated to match the EEG input size. For each model (with a specific seed), we determined the 99th percentile of gradient amplitudes across all 200 test samples and used this as the amplitude threshold for that model, resulting in five different thresholds. Finally, we computed a histogram showing the number of gradients exceeding their respective thresholds across time for all images and seeds (1,000 samples in total).

We extended this analysis to all participants and image encoders, with the results shown in Figures 8 to 10. The findings consistently indicate that human-aligned models focus more on earlier time windows, associated with low-level visual processing, compared to their unaligned counterparts, further corroborating the main paper's results.

Figure 11 illustrates the same procedure applied to two sample models (the top performers in the retrieval task) trained with *gLocal* image embeddings. The results indicate that there is no consistent difference in the time windows attended to by *gLocal* models compared to the original models.

Finally, we computed the gradients for models trained with *Harmonization* embeddings. Figure 12 and Figure 13 show histograms of large gradient values over time for four different architectures. Compared to their unaligned counterparts, the *Harmonized* models tend to exhibit larger gradients at later time points. Interestingly, this shift is more pronounced in the ConvNext and LEViT architectures (Figure 12), where human alignment was shown to degrade the model performance (Figure 3c).

## G.2. Spectral Analysis

To analyze the frequency contributions to the model's predictions, we computed the energy of the Fourier transform of the gradient maps. Following the same procedure as in the previous section, we extracted gradients from `encoder.0.projection.0`, retained only positive values, normalized them between 0 and 1, and interpolated them to match the EEG input size. The five frequency bands were defined as described in Section 5.5. We measured the power spectral density (PSD) using the periodogram method. The periodogram of a signal is obtained by squaring its Fourier transform values, as shown in Equation (3).

$$S_{xx}(f_k) = \frac{1}{N} |X_k|^2, \quad k = 0, 1, ..., N-1 \ , \tag{3}$$

where $X$ is the discrete Fourier transform of the signal and is computed as:

$$X_k = \sum_{n=0}^{N-1} x_n e^{-i2\pi kn/N} \tag{4}$$

The normalized energy of each frequency band was calculated by summing the power values within that band and dividing by the total signal energy (sum of power values across all frequencies). This process was performed separately for each seed, and the bars in Figure 4b and Figures 14-16 represent the mean and standard error across all seeds.

We repeated the procedure for different subjects and image encoders. As shown in Figures 14-16, models trained with human-aligned embeddings consistently exhibited greater focus on the Alpha and Beta bands, while original models emphasized the Delta band. This trend was observed across nearly all participants and image encoders. Although the Gamma band contributed more to predictions in some cases for human-aligned models, its effect was less consistent than the differences observed in other frequency bands.

Following the same approach, we repeated the analysis for the two best-performing *gLocal* models. The results, visualized in Figure 17, show that, in most cases, the original models attend more to Alpha and Beta frequencies compared to the *gLocal* models.

Similar to the previous section, we computed the normalized energy of the gradients in each EEG frequency band for the *Harmonized* models and compared them to their unaligned counterparts. Interestingly, for the LEViT and ConvNext

architectures, where the unaligned models outperform their human-aligned versions—the original models show higher energy in the alpha and beta bands (Figure 18). In contrast, for ResNet-50 and VGG16, where the human-aligned models achieve slightly better performance, the aligned models exhibit greater attention to the alpha and beta bands (Figure 19).

As discussed in the paper, we believe the difference between *gLocal* and *Dreamsim* stems from the datasets used to train them. The superior performance of *Dreamsim* in visual decoding from brain activity may be due to its focus on information that is emphasized by the experimental design used to collect brain data. In contrast, other models are optimized to highlight features such as categorical differences, which may not be well-captured in brain data collected using RSVP experiments.

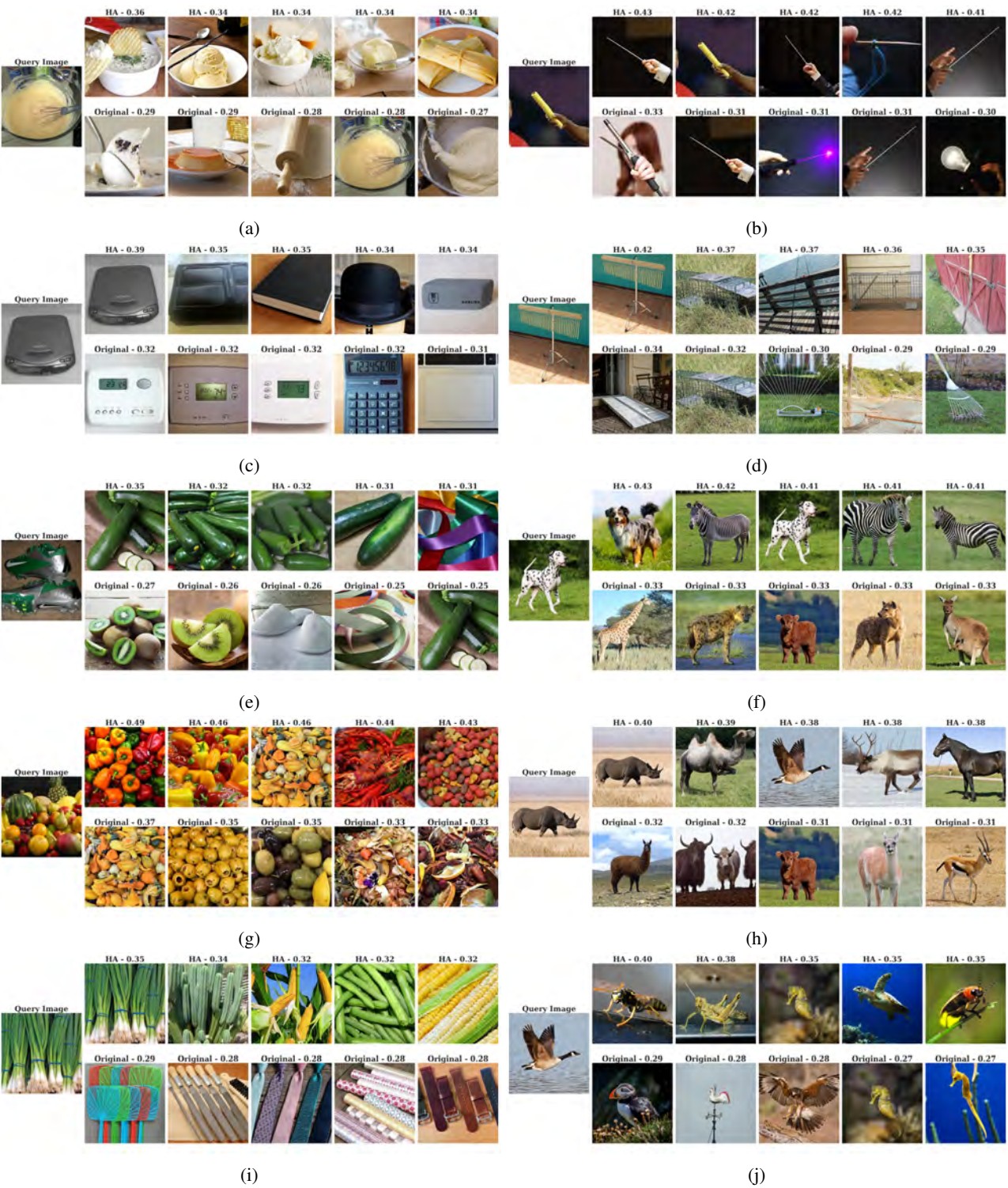

*Figure 6.* **Top-5 retrieved images from a larger image database (Dreamsim).** Query EEG signals were selected from the test set but the search for the corresponding image happened at the combined test and train image sets. Top rows show human-aligned (HA) and bottom rows show original models. The numbers are cosine similarity between the EEG and image embeddings. *Dreamsim* was used as the alignment method.

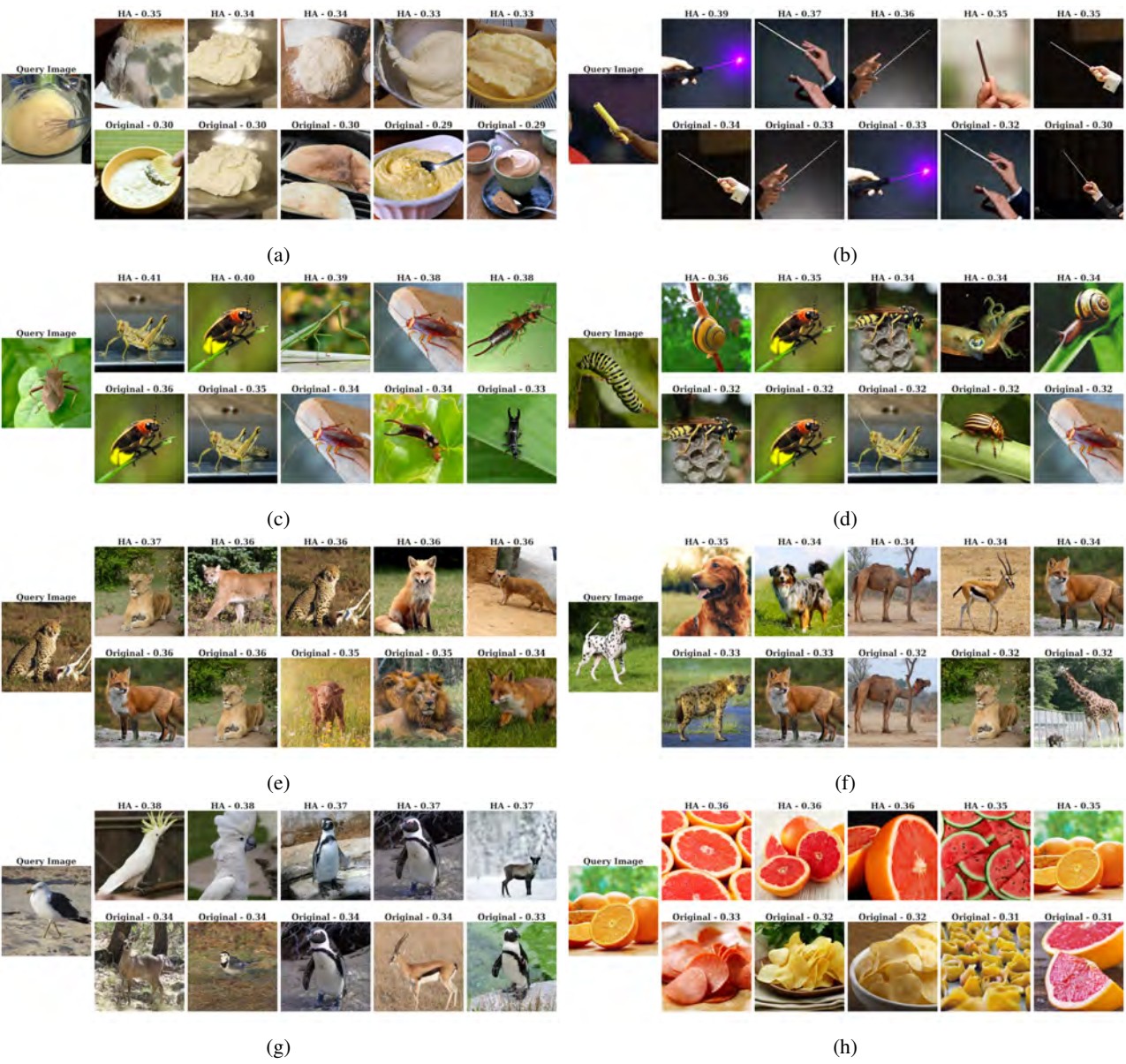

*Figure 7.* **Top-5 retrieved images from a larger image database (gLocal).** Query EEG signals were selected from the test set but the search for the corresponding image happened at the combined test and train image sets. Top rows show human-aligned (HA) and bottom rows show original models. The numbers are cosine similarity between the EEG and image embeddings.

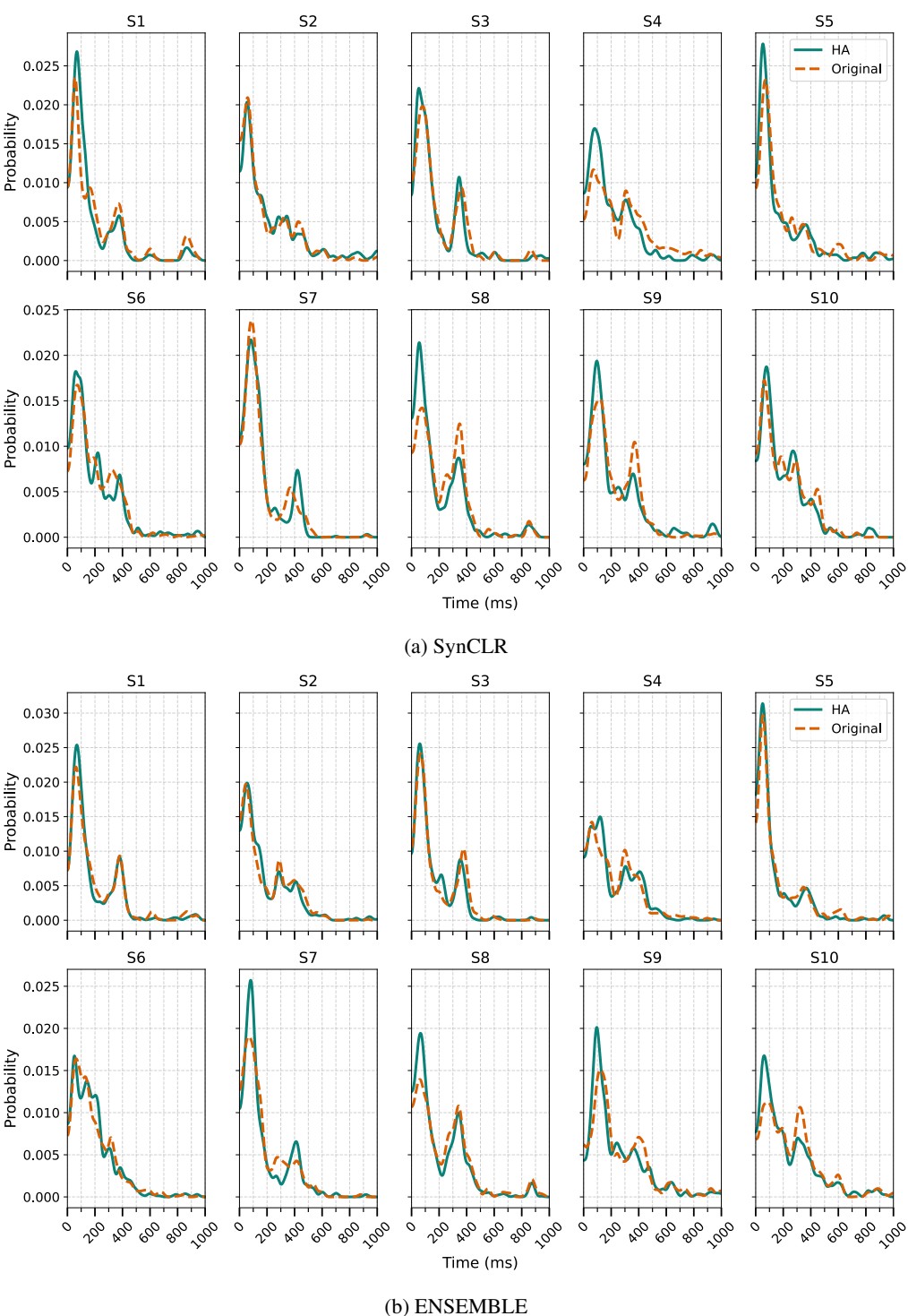

*Figure 8.* **Comparing gradient maps for EEG models–temporal analysis (Part 1).** EEG encoders are trained with original unaligned and human-aligned image embeddings using Dreamsim as the alignment method.

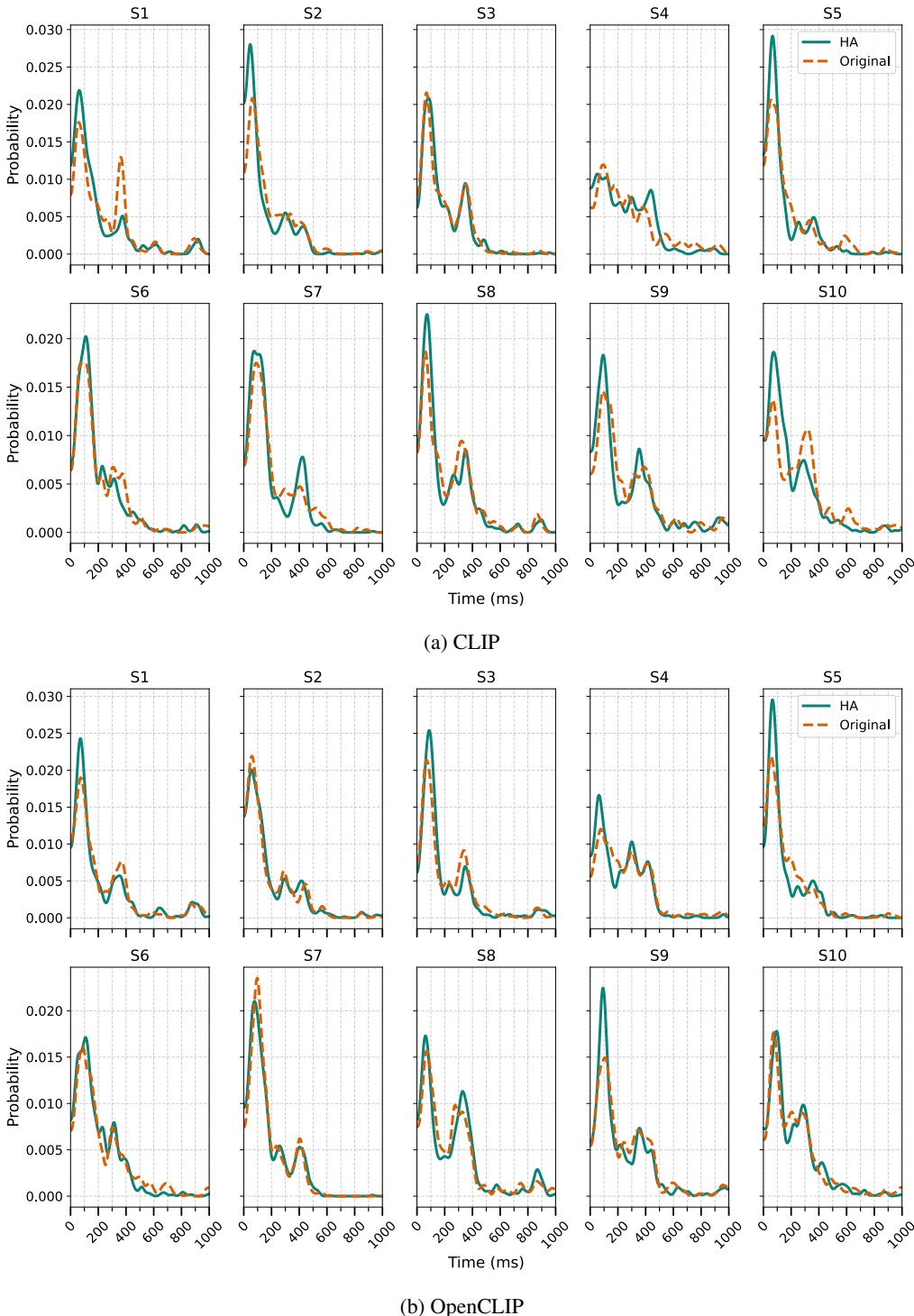

(a) CLIP

(b) OpenCLIP

*Figure 9.* **Comparing gradient maps for EEG models–temporal analysis (Part 2).** EEG encoders are trained with original unaligned and human-aligned image embeddings using Dreamsim as the alignment method.

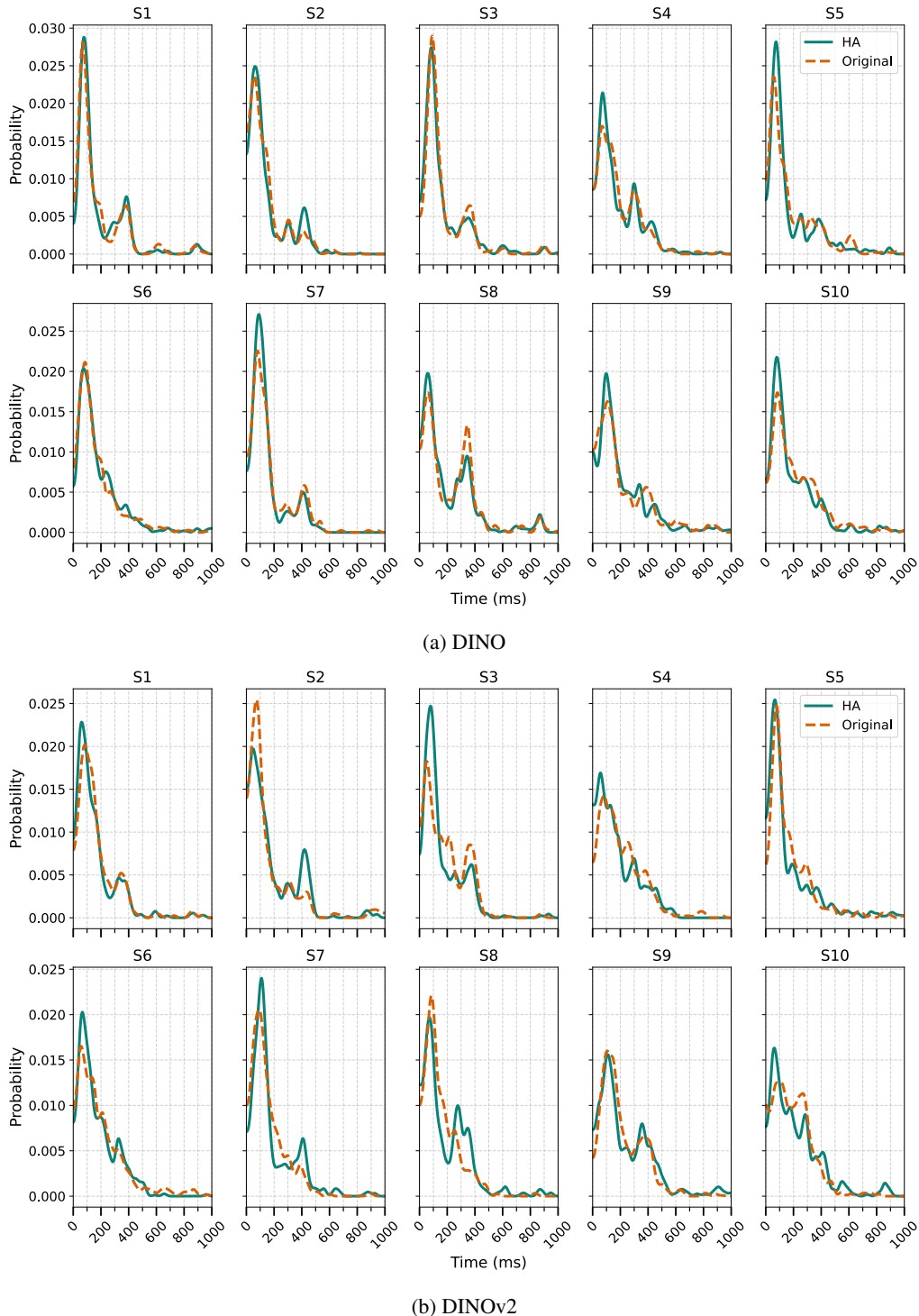

(a) DINO

(b) DINOv2

*Figure 10.* **Comparing gradient maps for EEG models–temporal analysis (Part 3).** EEG encoders are trained with original unaligned and human-aligned image embeddings using Dreamsim as the alignment method.

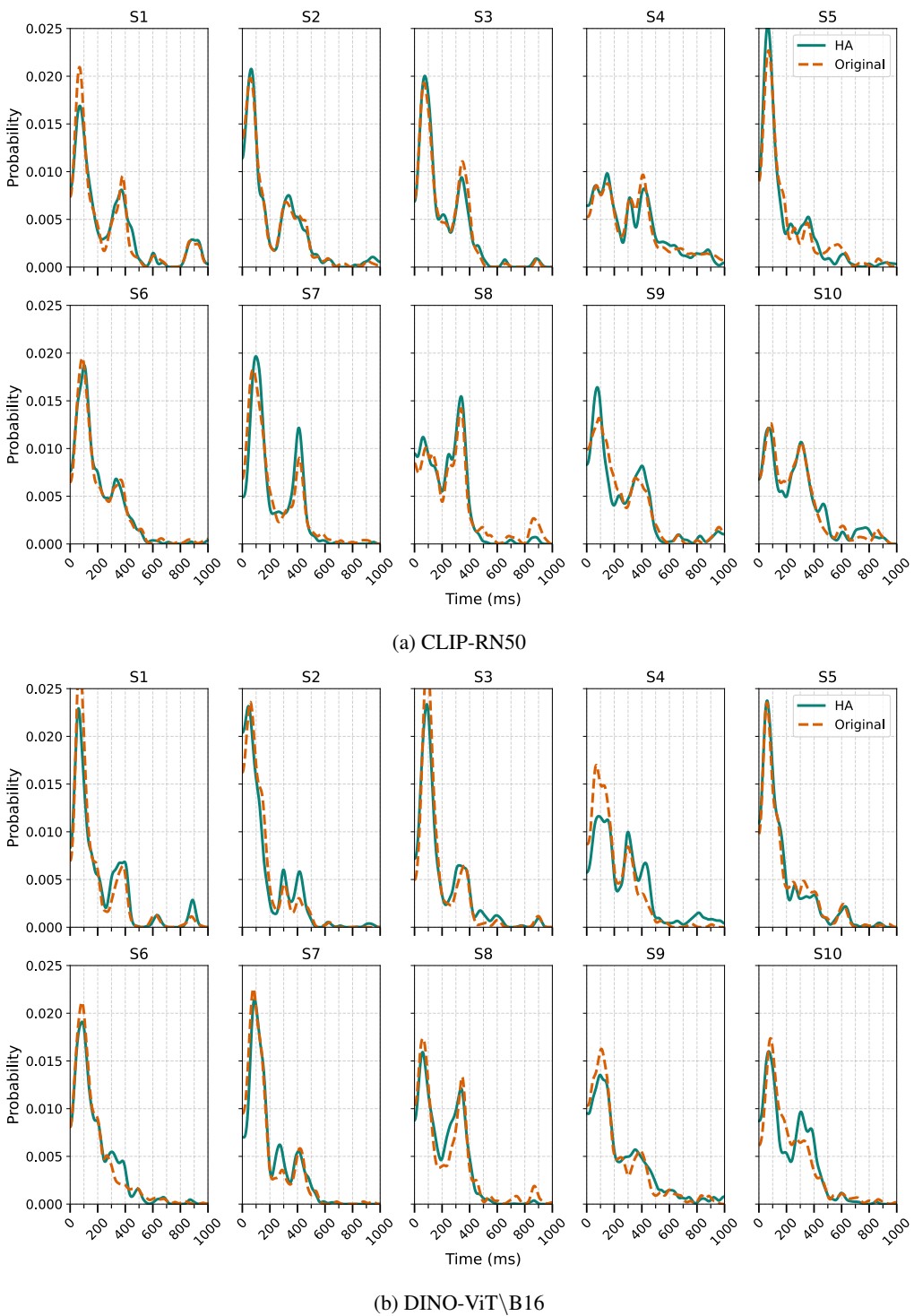

(a) CLIP-RN50

(b) DINO-ViT\B16

*Figure 11.* **Comparing gradient maps for EEG models–temporal analysis (gLocal).** EEG encoders are trained with original unaligned and human-aligned image embeddings using gLocal as the alignment method.

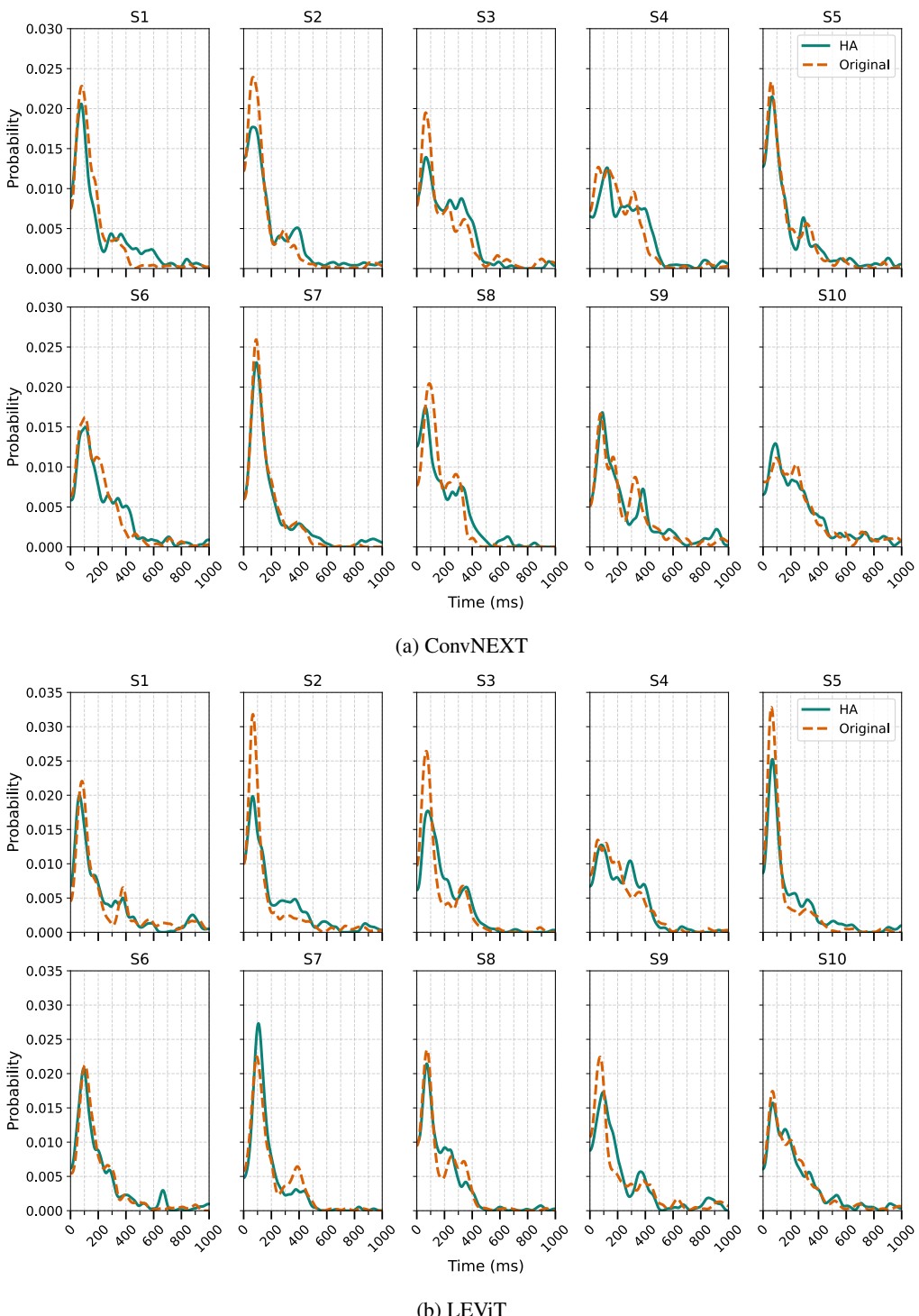

(a) ConvNEXT

(b) LEViT

*Figure 12.* **Comparing Gradient Maps for EEG Models–Temporal Analysis (Harmonization-Part 1).** EEG encoders are trained with original unaligned and human-aligned image embeddings using Harmonization as the alignment method.

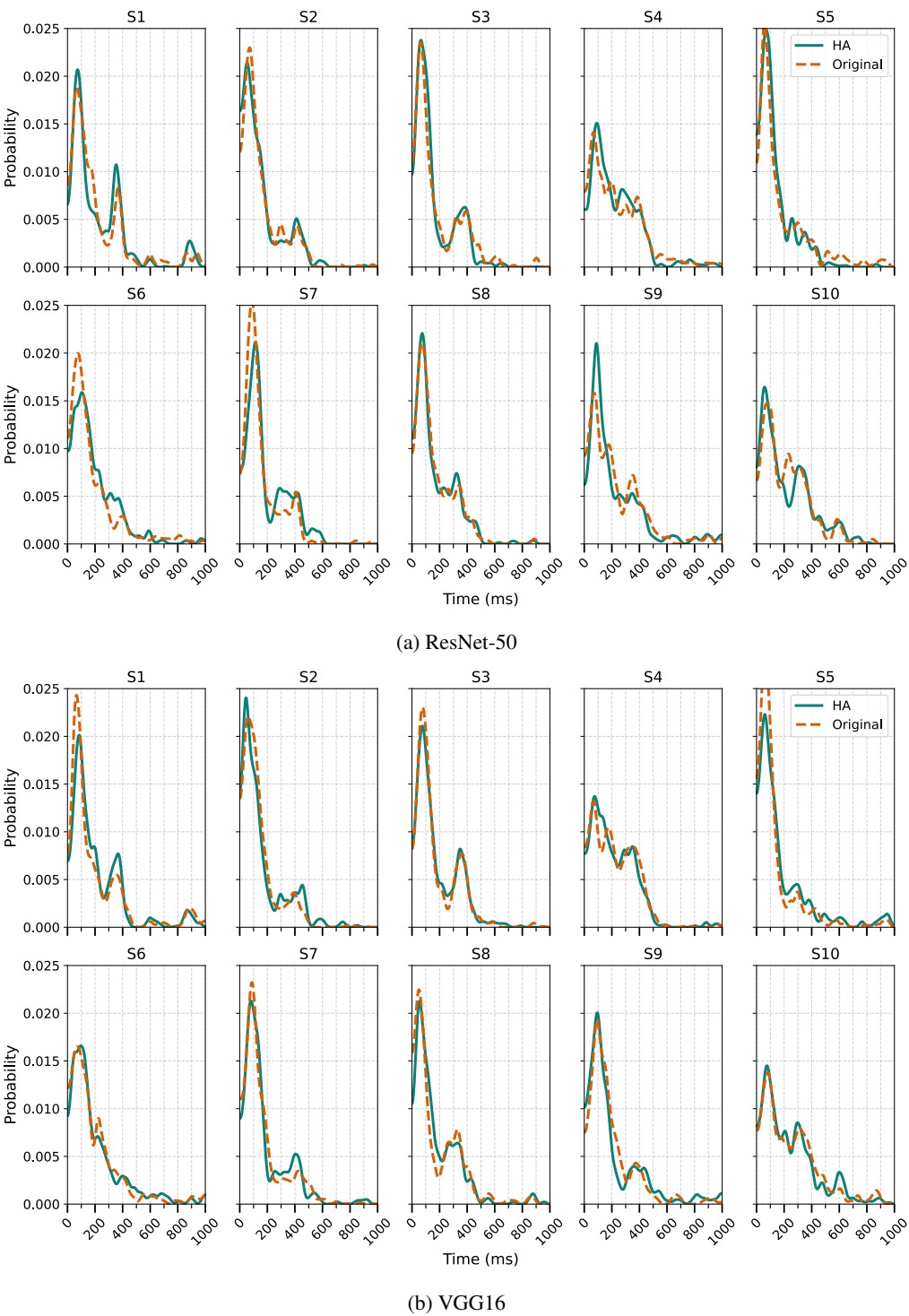

*Figure 13.* **Comparing gradient maps for EEG models–temporal analysis (Harmonization-Part 2).** EEG encoders are trained with original unaligned and human-aligned image embeddings using Harmonization as the alignment method.

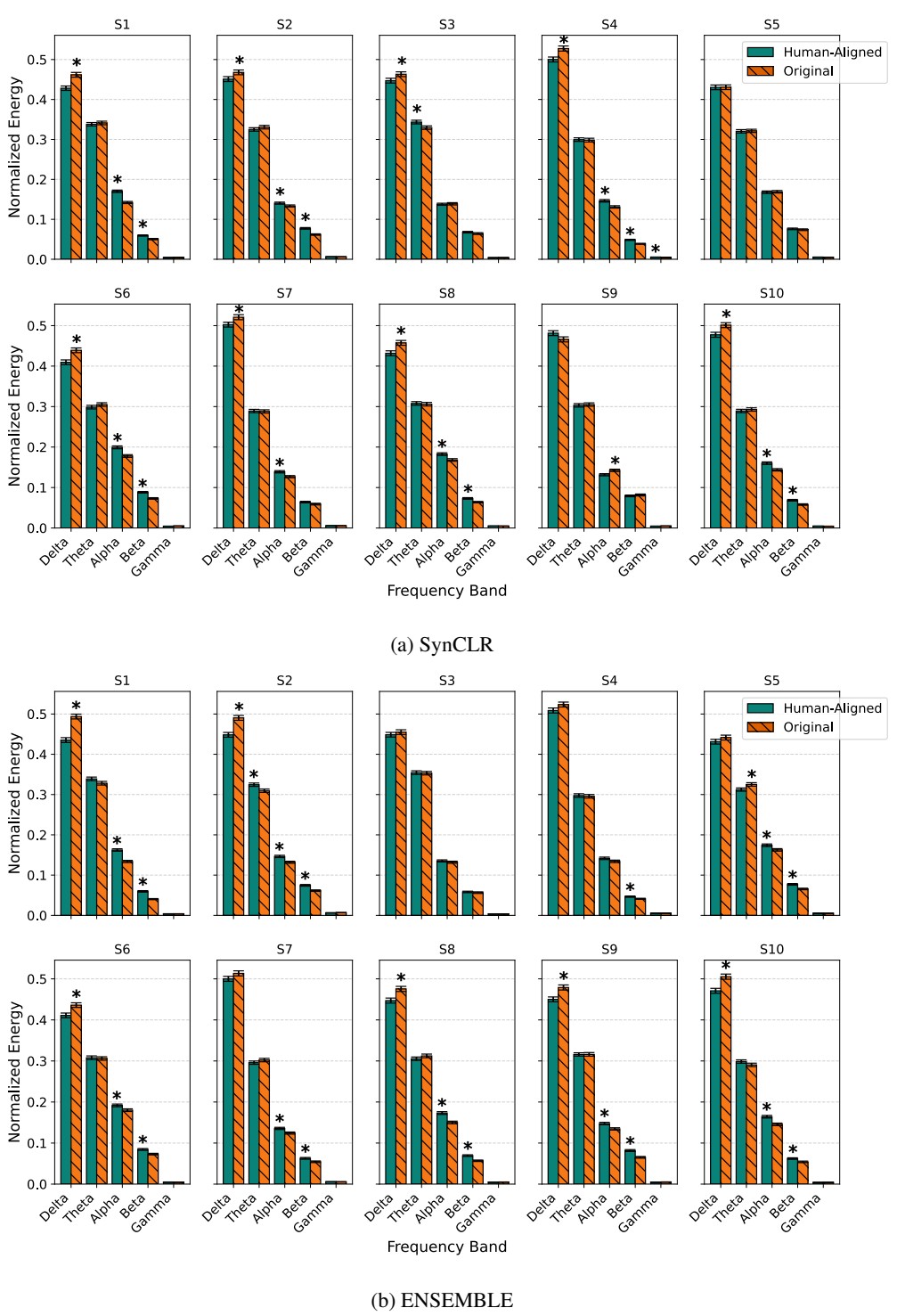

*Figure 14.* **Comparing gradient maps for EEG models–spectral analysis (Part 1).** EEG encoders are trained with original unaligned and human-aligned image embeddings using Dreamsim as the alignment method. Stars over bars represent a significant difference (greater) with $p < 0.05$.

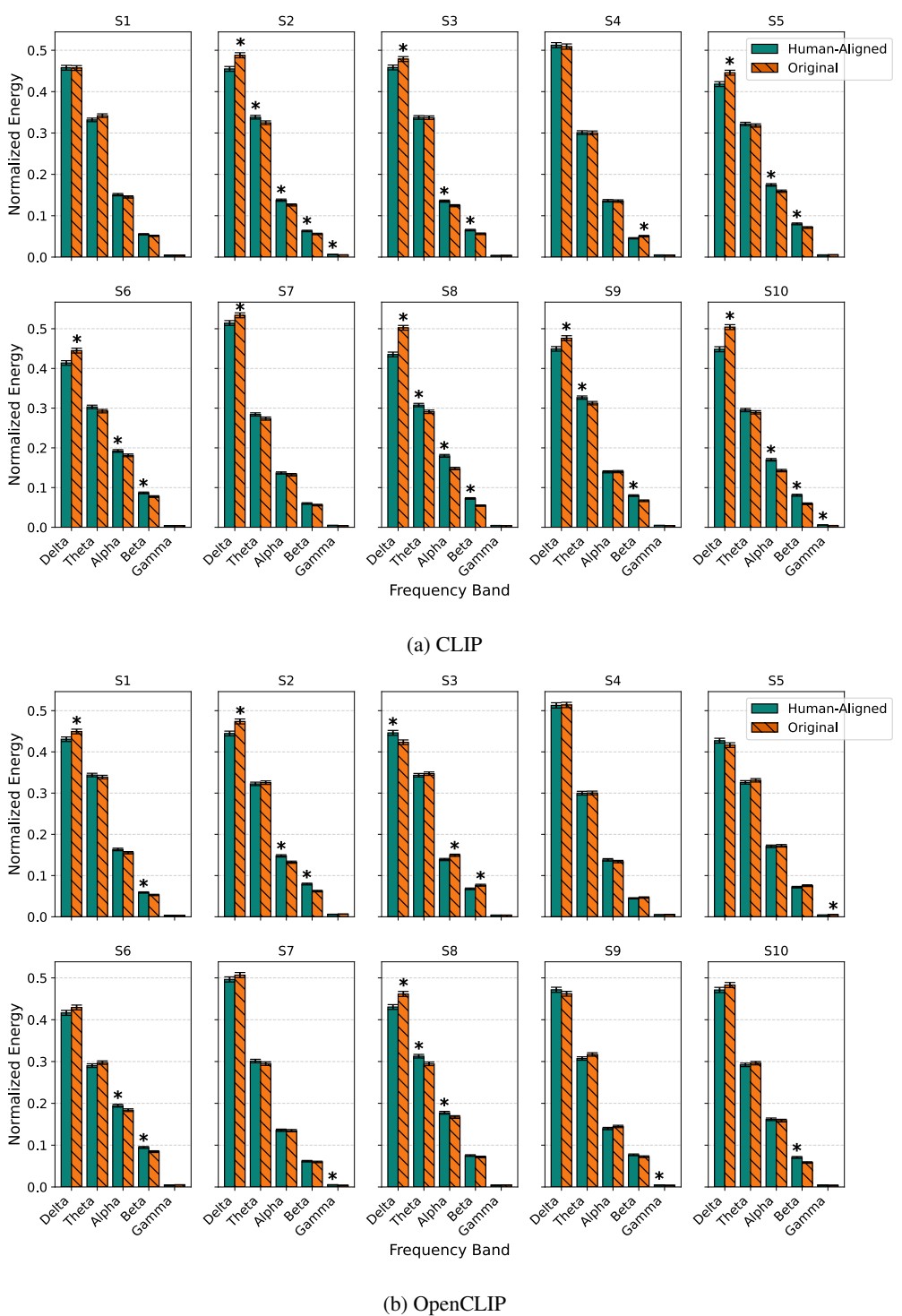

(a) CLIP

(b) OpenCLIP

*Figure 15.* **Comparing gradient maps for EEG models–spectral analysis (Part 2).** EEG encoders are trained with original unaligned and human-aligned image embeddings using Dreamsim as the alignment method. Stars over bars represent a significant difference (greater) with $p < 0.05$.

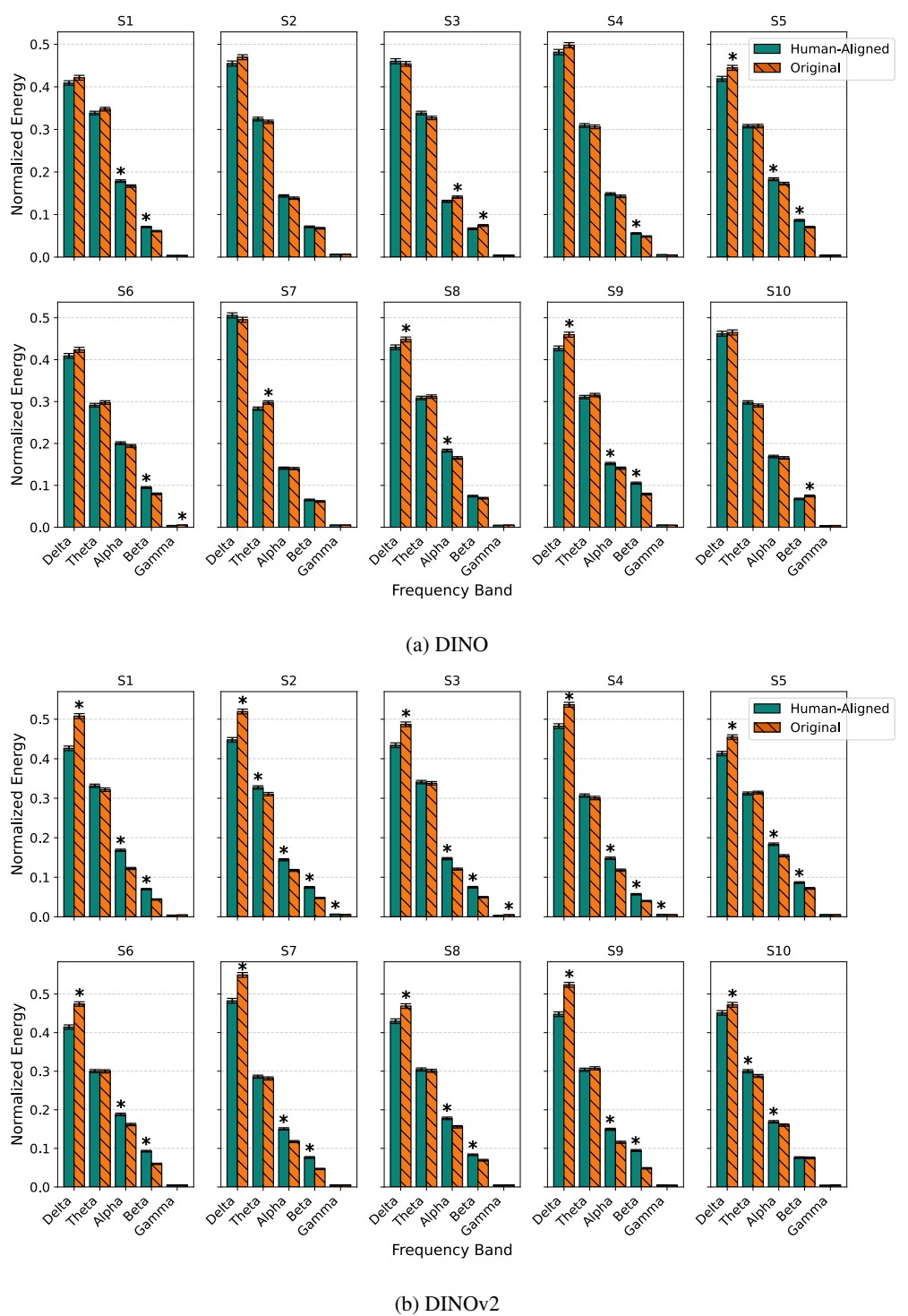

(a) DINO

(b) DINOv2

*Figure 16.* **Comparing gradient maps for EEG models–spectral analysis (Part 3).** EEG encoders are trained with original unaligned and human-aligned image embeddings using Dreamsim as the alignment method. Stars over bars represent a significant difference (greater) with $p < 0.05$.

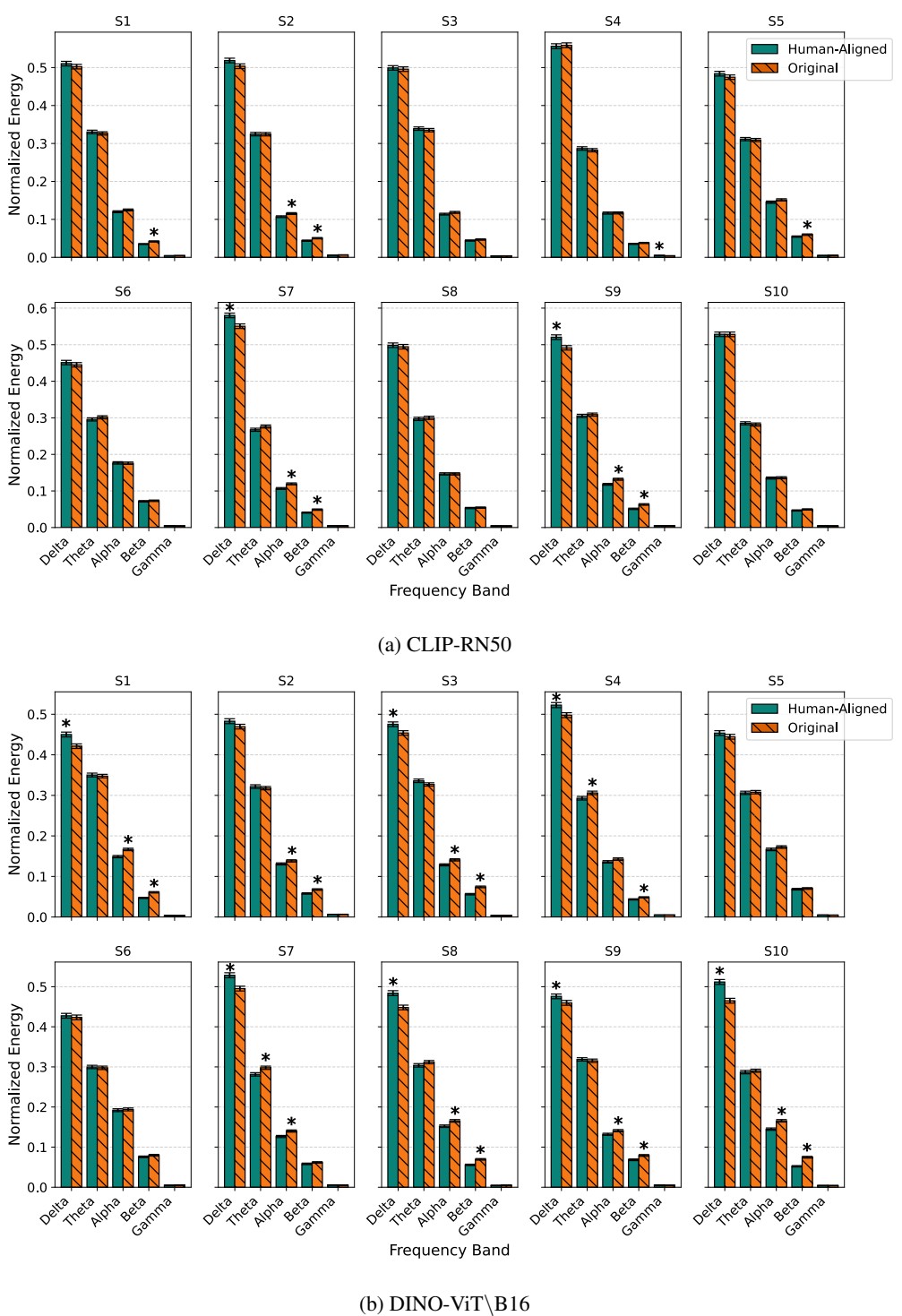

(a) CLIP-RN50

(b) DINO-ViT\B16

*Figure 17.* **Comparing gradient maps for EEG models–spectral analysis (*gLocal*).** EEG encoders are trained with original unaligned and human-aligned image embeddings using *gLocal* as the alignment method. Stars over bars represent a significant difference (greater) with $p < 0.05$.

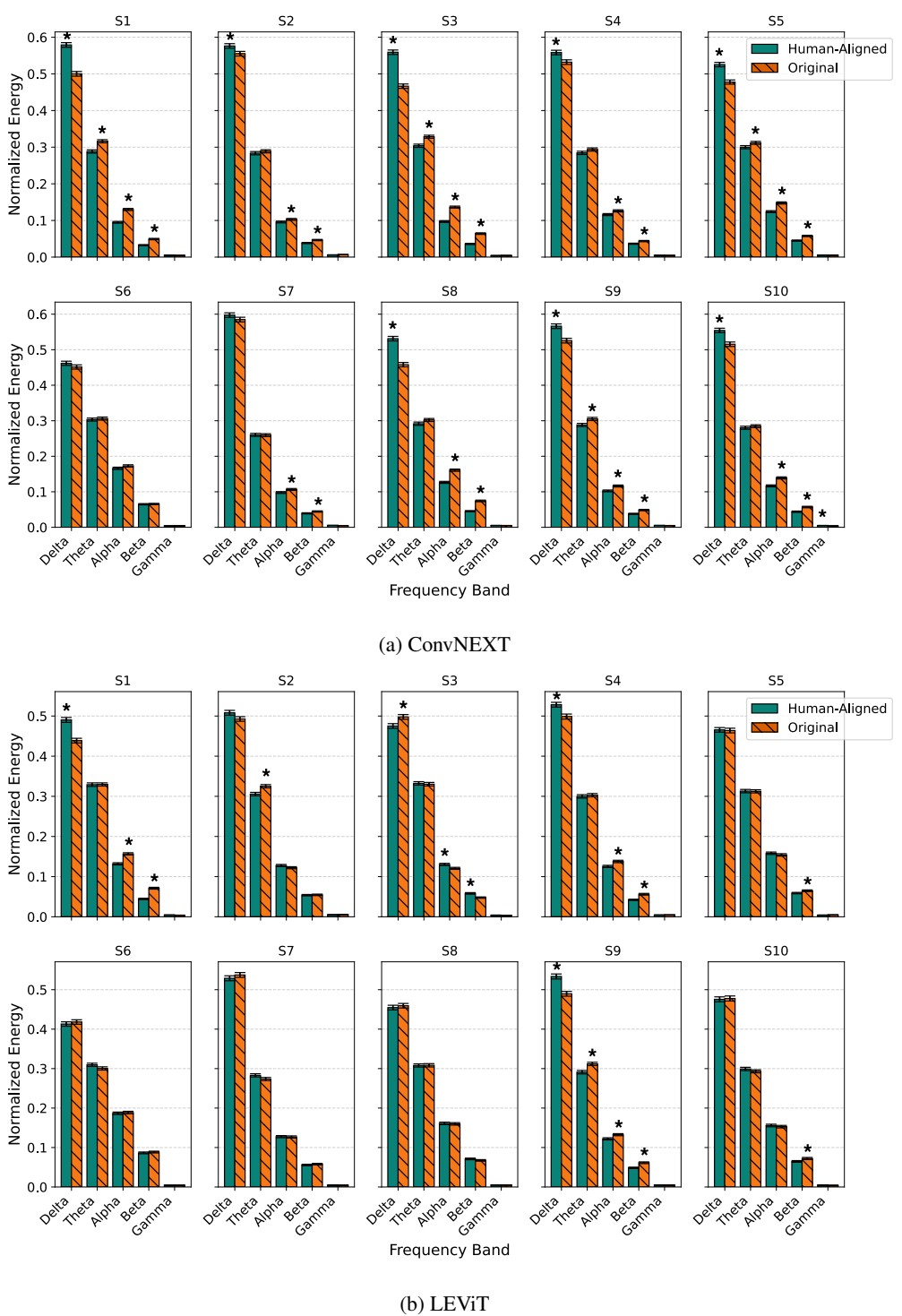

(a) ConvNEXT

(b) LEViT

*Figure 18.* **Comparing gradient maps for EEG models–spectral analysis (*Harmonization*-Part 1).** EEG encoders are trained with original unaligned and human-aligned image embeddings using *Harmonization* as the alignment method. Stars over bars represent a significant difference (greater) with $p < 0.05$.

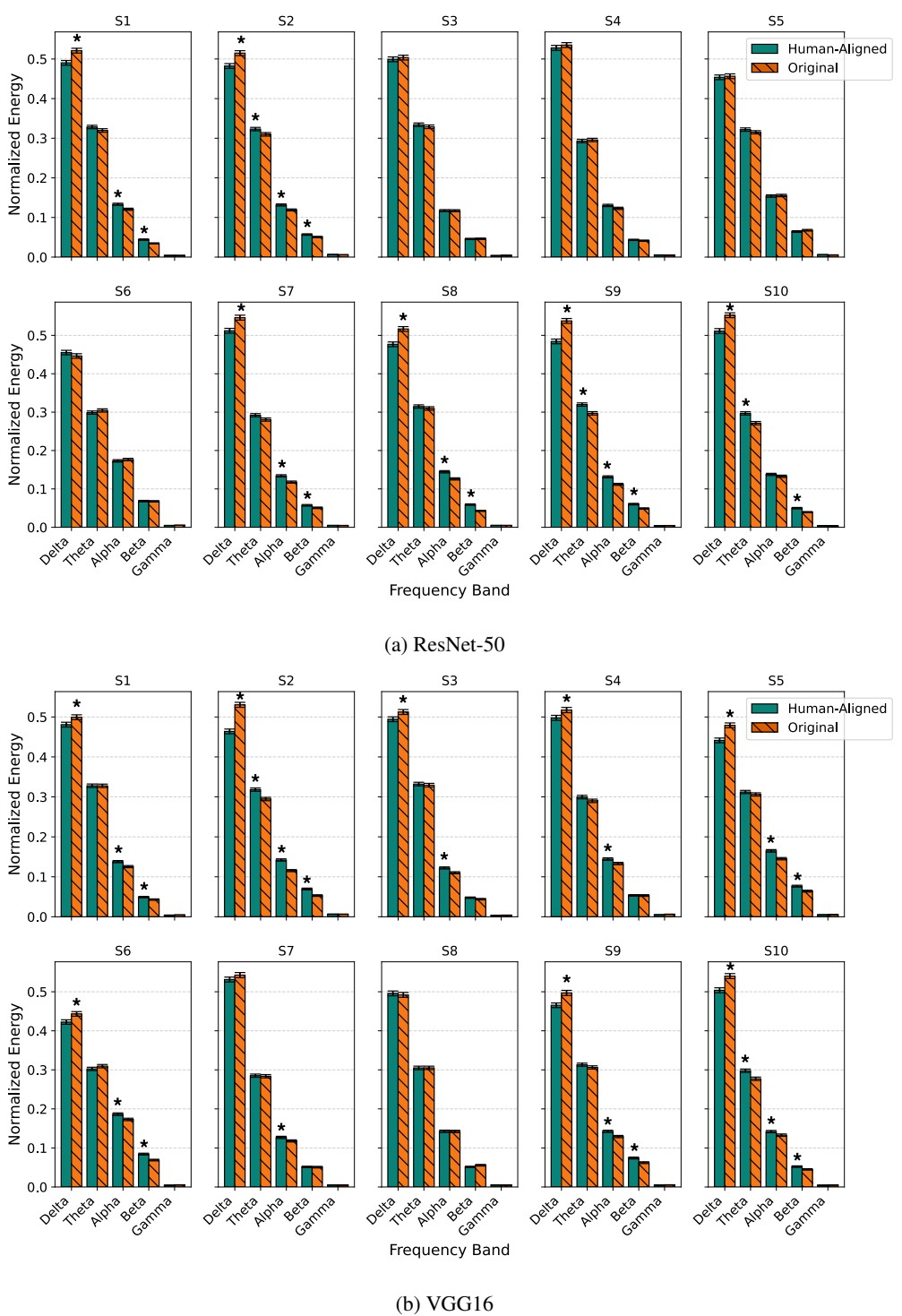

(a) ResNet-50

(b) VGG16

*Figure 19.* **Comparing gradient maps for EEG models–spectral analysis (*Harmonization*-Part 2).** EEG encoders are trained with original unaligned and human-aligned image embeddings using *Harmonization* as the alignment method. Stars over bars represent a significant difference (greater) with $p < 0.05$.

