# OpenReview forum: "Human-Aligned Image Models Improve Visual Decoding from the Brain"
_ICML.cc/2025/Conference — ICML 2025 poster_

### Official Review · Reviewer_KJCA · 2025-03-09

**Overall Recommendation:** 3

**Summary:**

The authors tackle the task of image decoding from brain activity (EEG and MEG). On this task, most of the existing papers map brain data to pretrained vision encoders such as CLIP and DINO. The authors propose to analyze the role of these vision encoders and show that aligning them with human perception boosts performance on image retrieval tasks. They conduct experiments by analyzing different types of neural architectures for mapping brain data to vision embeddings and provide a biological analysis showing the representation learned by these human-aligned encoders have more meaningful spectral and temporal response compared to non-aligned ones.

**Claims And Evidence:**

The claims are clear.

**Essential References Not Discussed:**

No

**Experimental Designs Or Analyses:**

The experimental part is well designed but would benefit from additional experimental results to validate the claim stating that human-aligned models are better at decoding.
- Evaluation on more tasks. The authors only evaluate models on image retrieval. it would be interesting to see if it generalizes to other vision tasks such as image generation, captioning.
- Evaluation on other brain modalities. The authors choose to focus on MEG and EEG data using THINGS datasets. Since the fMRI modality is also available in THINGS dataset and is commonly used for image decoding, it would be valuable to examine the impact of human-aligned encoders on fMRI data to determine if the findings are also applicable to this modality too.

**Methods And Evaluation Criteria:**

The method makes sense and is well justified.

**Other Comments Or Suggestions:**

No

**Other Strengths And Weaknesses:**

The paper is well written and easy to follow. It presents interesting findings by showing that human-aligned image encoders enhance the performance of image retrieval from brain signals, opening up new avenues for future research.

**Questions For Authors:**

The results related to the generalization to other subjects in Section 5.3 are not very clear. Indeed, could you please elaborate on why the generalization would work for CLIP and openCLIP and not DINO network ?

**Relation To Broader Scientific Literature:**

The paper focuses on image decoding from brain activity, a field that has seen significant advancements in recent years due to the emergence of large pretrained vision models. Most existing studies involve aligning brain representations with image encoders like CLIP which was shown to not align well with human perception. This paper demonstrates that using an image encoder better aligned with human similarity judgments yields a better decoding performance on retrieval tasks.

**Theoretical Claims:**

No proofs

---

> ### Author Rebuttal · Authors · 2025-03-28
>
> We thank Reviewer KJCA for their constructive review and insightful suggestions. We agree that evaluating the impact of human-aligned embeddings across a broader range of vision tasks would be highly valuable. In this work, we focused on image retrieval as a direct and interpretable proxy for brain-image alignment. Nonetheless, we see strong potential for extending this framework to other tasks that rely on shared embedding spaces. As part of our ongoing work, we are exploring image reconstruction and plan to investigate whether human-aligned embeddings can also enhance brain-to-text mappings.
>
> Regarding your suggestion on incorporating fMRI data, we fully agree. In response, we have conducted additional experiments using the Natural Scenes Dataset (NSD) which has been commonly used in similar studies. The results—provided below—show a consistent improvement in retrieval performance when using human-aligned embeddings over unaligned ones, further supporting the generalizability of our findings.
>
> |       | CLIP     |          | DINOv2   |          | DINO     |          | ENSEMBLE |          | OpenCLIP |          | SynCLR   |          |
> |-------|----------|----------|----------|----------|----------|----------|----------|----------|----------|----------|----------|----------|
> |       | HA       | Base     | HA       | Base     | HA       | Base     | HA       | Base     | HA       | Base     | HA       | Base     |
> | top-1 | **45.0±4.3** | 23.8±2.4 | **45.3±3.6** | 22.5±2.1 | **49.1±3.7** | 45.9±3.7 | **54.6±4.3** | 48.1±4.0 | **49.5±4.3** | 35.4±3.8 | **58.2±4.6** | 48.2±4.3 |
> | top-5 | **76.6±4.2** | 53.8±4.1 | **75.5±3.5** | 49.1±3.2 | **79.7±3.7** | 76.8±3.6 | **83.4±3.7** | 79.6±3.4 | **79.2±4.1** | 67.8±4.3 | **86.7±3.6** | 78.7±3.1 |
>
> **Regarding your question:**
>
> Thank you for raising this question. Interestingly, while the original (unaligned) performance of models like DINO is comparable to that of human-aligned CLIP and OpenCLIP in the cross-subject setting, human alignment improves performance only for the latter. One possible explanation is that CLIP and OpenCLIP embeddings, due to their language supervision, have more semantically structured representations that benefit from additional alignment with human similarity judgments. In contrast, DINO and SynCLR—trained with self-supervised visual objectives—may already be well-matched to the structure of EEG signals in a way that does not benefit further from human alignment in the cross-subject setting. This suggests that the effectiveness of human alignment in promoting generalization may depend on the underlying inductive biases of the image encoder.

---

### Official Review · Reviewer_q6J2 · 2025-03-13

**Overall Recommendation:** 3

**Summary:**

This paper explores the problem of decoding visual images from the brain by replacing the visual encoder with a human-aligned visual encoder. The experiments demonstrate that the use of human-aligned visual encoder effectively improves the brain-image retrieval performance.

**Claims And Evidence:**

See Other Strengths And Weaknesses.

**Essential References Not Discussed:**

No.

**Experimental Designs Or Analyses:**

See Other Strengths And Weaknesses.

**Methods And Evaluation Criteria:**

Yes.

**Other Comments Or Suggestions:**

See Other Strengths And Weaknesses.

**Other Strengths And Weaknesses:**

Strengths

- This paper is well-written and easy to follow.
- This paper conducts extensive experiments across various EEG and image encoders, demonstrating the effectiveness of the use of human-aligned image encoders.
- This paper reveals an interesting finding that human-aligned encoders demonstrate greater compatibility with brain visual decoding tasks.

Weaknesses

- This paper focuses only on image retrieval task. However, brain decoding has another important task, reconstructing images from brain signals, which is more challenging and important. Extending the experiments to include the reconstruction task would provide a more convincing demonstration of the method’s effectiveness.
- The novelty is limited. Although effective, the model’s design and alignment strategy is quite simple, i.e., using the human-aligned models to align with EEG features through contrastive loss.
- The visualization results in Fig. 4 cannot fully support the biological interpretation in Section 5.5. According to Fig. 4 (b), the PSD difference of human-aligned models and original models is trivial，making it difficult to conclude that “Models trained with original embeddings focused more on Delta, while human-aligned models emphasized Alpha, Beta, and Gamma”. Additionally, in Fig. 4(c), EEG features aligned using different methods appear to focus on nearly identical brain regions, which does not sufficiently explain why the human-aligned models perform better. Further visualization across more models may be necessary to provide stronger evidence for these claims.

**Questions For Authors:**

See Other Strengths And Weaknesses.

**Relation To Broader Scientific Literature:**

See Other Strengths And Weaknesses.

**Theoretical Claims:**

See Other Strengths And Weaknesses.

---

> ### Author Rebuttal · Authors · 2025-03-28
>
> We thank Reviewer q6J2 for their thoughtful review. Please find our responses to your concerns and suggestions below.
>
> **Regarding the image reconstruction task:**
>
> Thank you for highlighting this important point. We agree that image reconstruction is a key aspect of brain decoding and would be an important addition to our retrieval results. Our current focus is on retrieval as it provides a direct measure of alignment between brain activity and image embeddings—our primary objective in this work. That said, we recognize the significance of reconstruction and are exploring it in our current work.
>
> State-of-the-art reconstruction methods typically use pretrained diffusion models based on latents from the CLIP representation space and project brain activity to that using an alignment loss such as contrastive or MSE loss (Scotti et al. 2024, Li et al. 2024, Benchetrit et al. 2024). However, a challenge is that current generative models are trained on original CLIP embeddings, so using human-aligned embeddings would require retraining or fine-tuning the generative model. This task is broader than brain-image alignment and primarily tests the general utility of human-aligned embeddings as a latent space for generative diffusion models. We thought our setup provided a more direct and easier-to-evaluate scenario that allows us to isolate the effect of human alignment better, and that is why it was our focus.
>
> An alternative, inspired by MindEye, decouples retrieval and reconstruction. In our case, we can have two separate representation spaces: In one, brain signals are aligned with human-aligned embeddings for retrieval and with the original CLIP embeddings for reconstruction. We are also experimenting with a two-stage pipeline: brain signals are first mapped to the human-aligned space (for retrieval) and then further projected into the original CLIP space for reconstruction. This framework is implemented and under active evaluation. Preliminary results are promising, but further experimentation is needed to draw firm conclusions.
>
> We hope our ongoing work and future directions help address your concern regarding the broader impact of the study.
>
> **Regarding the novelty:**
>
> Thank you for the feedback. While the model design is intentionally simple, we believe the contribution lies in the insight that brain signals are more aligned with image embeddings that have been aligned with human similarity judgments. This alignment raises the interesting question of whether the representations of currently used models are suitable for brain signals. Also, it opens new doors to further investigate the connection between human-alignment and brain signal decoding. To our knowledge, this is the first work to systematically evaluate the impact of human-aligned representations on brain-based image retrieval across multiple modalities, models, and datasets. We hope this finding encourages future work to explore cognitively grounded embeddings in brain decoding.
>
> **Regarding the biological interpretations:**
>
> Thank you for this important observation. We agree that the visualizations should be interpreted with caution. As noted in the paper, we performed statistical tests to assess the significance of the differences and only reported effects that met the significance threshold. However, we acknowledge that these differences—while statistically supported—are subtle and should be viewed as suggestive evidence rather than definitive explanations.
>
> Our intent was to provide preliminary insight into how human alignment might influence model attention across spectral and spatial domains. We agree that more extensive visualizations across additional models and datasets would help strengthen these interpretations. While additional visualizations for other models and participants are included in Appendix F2, we will revise the text to better highlight the limitations of these findings.

---

> > ### Comment · Reviewer_q6J2 · 2025-04-07
> >
> > After reading author's response, I decide to raise my rating. Although the method design is quite simple, the finding that brain signals are more aligned with human-aligned visual embeddings is interesting.

---

### Official Review · Reviewer_9vzB · 2025-03-15

**Overall Recommendation:** 3

**Summary:**

This paper compares the image identification (decoding) performance of human-aligned image embedding models and their unaligned counterparts. The authors find that human-aligned models generally performed better than the unaligned models in EEG and MEG, after evaluating several different base image encoders and brain signal encoders. The authors also claim that the gradients of the EEG encoder more closely match human perception when using human-aligned image embeddings yields.

**Claims And Evidence:**

1. The authors claim that Dreamsim performs better than the two alignment alternatives (gLocal and Harmonization) and go on to speculate the cause. But, the image encoders used to measure improvement are not the same for each alignment method, so their results are not directly comparable. In Figure 3, it is also unclear which image encoders _do_ have direct comparisons between alignment methods (i.e., which encoders are both in 3a and 3b).

**Essential References Not Discussed:**

Unless the authors are specifically discussing previous efforts to decode still images in Lines 154-161 (which I believe they should not), discussion of Nishimoto et al. 2011 paper belongs in Section 3.1: https://www.sciencedirect.com/science/article/pii/S0960982211009377

**Experimental Designs Or Analyses:**

The analyses seem sound -- in particular, the authors are explicit about which concepts are distinct to which split of the EEG and MEG datasets. I do have a concern about an overlap in concepts in the human alignment datasets and the brain datasets. (I ask about this in the Questions section as well.)

**Methods And Evaluation Criteria:**

The authors use retrieval to evaluate the quality of their decoder.
1. I believe some of the specifics about the retrieval process (like which images are made available to the decoder) are not stated.
2. Further, I believe a task like open domain reconstruction (rather than selecting from a known set of candidates) would be more interesting, albeit significantly more computationally expensive. I think this because the test set, unlike the training set, seems to contain 200 concepts each with a single unique image. This means that the model might not need to learn to (for example) differentiate between viewpoints of a single object -- simply differentiating something like texture or color may be sufficient.

**Other Comments Or Suggestions:**

1. In Figure 3a, consider reordering the bars to match the order of the models in Tables 1-3.

**Other Strengths And Weaknesses:**

Other strengths:
1. The authors are thorough and trained a wide combination of EEG encoders and image encoders.

**Questions For Authors:**

1. Were any of the concepts in the EEG and MEG datasets also present in the human alignment datasets (e.g. NIGHTS)? In particular were they part of the test split of the brain datasets?
2. At test time, which concepts/images is the encoder given access to? Only the 200 images associated with the 200 test concepts, or also the images/concepts from the training set? (If I missed this in the paper, please point me to where it was discussed.)
2. In Table 1, the base performance of DINOv2 is substantially worse than DINO, though the gap narrows after human alignment. What could be the cause of this?
3. In Table 2, the Ensemble and DINO models show no improvement after HA. What makes cross-subject decoding so different from within-subject decoding to cause this?
4. In Table 2, S1 shows a catastrophic drop in performance for OpenCLIP after HA. What is going on there? It doesn't seem to just be a bad seed.
5. Around Line 408 in column 2, the authors speculate that the datasets for Harmonization emphasize features that "require careful inspection and attention", while the RSVP setup of the experiments emphasize the recognition of more low-/mid-level features. This would be borne out by looking at the temporal gradients -- is a shift toward later timepoints actually seen in the data? (In the appendix I saw some gradient maps for gLocal but not Harmonization.)

**Relation To Broader Scientific Literature:**

This work shows that human-alignment efforts in machine learning may have deeper impacts in modeling (decoding) brain processes.

**Theoretical Claims:**

No theoretical claims were made in this paper.

---

> ### Author Rebuttal · Authors · 2025-03-28
>
> We thank Reviewer 9vzB for their careful review and constructive suggestions. We appreciate your thoughtful engagement with our work.
>
> Your concern about the image encoders differing across alignment methods is valid. However, we would like to clarify that our comparisons are always made within the same image encoder, specifically, between its original embeddings and its human-aligned embeddings. Our goal is to assess whether human alignment improves brain-image alignment for a given encoder, not to compare across encoder architectures. The inclusion of multiple image encoders was intended to test the consistency of this effect across different visual representation models. Thank you also for suggesting the inclusion of Nishimoto et al., 2011. We agree it is a relevant and influential study, and we will incorporate it into the related work section of the revised paper. Below, we address your specific questions in detail.
>
> **Question 1:**
>
> Thank you for this important question. Based on Fu et al. (2023), the THINGS categories used in the THINGS-EEG and THINGS-MEG datasets are not directly included in the NIGHTS dataset. The Harmonization component of NIGHTS is based on ImageNet-derived datasets. As for gLocal, it uses odd-one-out triplets from the same THINGS database, so there is likely some concept overlap between its training data and the EEG/MEG test sets.
>
> However, we would like to emphasize two points. First, despite potential concept overlap, gLocal-aligned models do not outperform unaligned models on EEG/MEG retrieval, suggesting that overlap alone does not explain performance differences. Second, these concepts are already likely present in the pretraining datasets of the original image encoders, which are large-scale vision models trained on broad and diverse datasets. Therefore, any overlap applies equally to both human-aligned and unaligned models and does not undermine the claims of the paper.
>
> **Question 2:**
>
> In the main results, the encoder only has access to the 200 test concepts and their associated images during retrieval. We will clarify this more explicitly in the revised text. For completeness, we also evaluated retrieval on a larger image database—including both training and test concepts—and reported those results in Appendix E. Notably, models trained with human-aligned embeddings continue to outperform the original models in this more challenging setting.
>
> **Question 3:**
>
> One possible explanation is that DINOv2 embeddings may capture different visual features or emphasize different inductive biases compared to DINO, leading to a less optimal alignment with EEG representations in the unaligned setting. After human alignment, however, both models are optimized toward human perceptual similarity, which reduces the gap by bringing their representations closer to behaviorally relevant dimensions. This suggests that human alignment can partially mitigate architectural differences by enforcing a shared similarity structure.
>
> **Question 4:**
>
> Cross-subject decoding introduces additional variability due to individual differences in neural representations, which may overshadow the benefits of human alignment. In this setting, the EEG encoder must generalize across subjects, making it more difficult to exploit the finer-grained structure introduced by human alignment.
>
> **Question 5:**
>
> Thank you for catching this. The reported value is a typo—the correct performance for OpenCLIP with HA on S1 is **11.5 ± 0.55**. As noted in the paper, all results are averaged over five random seeds to reduce sensitivity to a specific seed. We will correct this in the revision.
>
> **Question 6:**
>
> You are right; we only included temporal gradient maps for Dreamsim and gLocal in the original submission. Based on your suggestion, we computed the gradient distributions over time for models trained with Harmonization-aligned embeddings and will include them in Appendix F1 of the revised paper. As anticipated, we do not observe a consistent shift toward earlier time points compared to the unaligned models. This suggests that Harmonization-aligned features do not rely more heavily on early, low-level EEG responses, and the alignment may not strongly alter temporal emphasis in this context.

---

> > ### Comment · Reviewer_9vzB · 2025-04-04
> >
> > Thank you for your thorough responses. I've increased my score.

---

### Official Review · Reviewer_ewMg · 2025-03-17

**Overall Recommendation:** 3

**Summary:**

The paper explores the application of human-aligned image encoders to enhance the decoding of visual information from brain signals, specifically EEG and MEG data. The authors propose that image encoders fine-tuned to align with human perceptual similarity judgments improve the mapping of brain activity to visual stimuli compared to standard pre-trained encoders. The main findings indicate that integrating human-aligned image encoders, such as Dreamsim, into a brain-to-image decoding framework increases image retrieval accuracy. The study employs a contrastive learning approach, aligning brain signal embeddings with image embeddings in a shared latent space using the InfoNCE loss. Comprehensive experiments across various EEG architectures, image encoders, and brain imaging modalities demonstrate consistent performance improvements. The authors also provide biological insights, showing that human-aligned models focus on early visual processing features and frequency bands linked to visual perception.

**Claims And Evidence:**

The primary claim—that human-aligned image encoders improve visual decoding from brain signals—is supported by extensive empirical evidence. The authors present quantitative results, such as top-1 and top-5 retrieval accuracies, across multiple participants, image encoders, and EEG architectures, showing consistent improvements with human-aligned models. The 21% improvement over state-of-the-art (from 28% to 62% top-1 accuracy with NICE and Ensemble models) is convincingly demonstrated in Figure 1 and Table 1. Additional evidence from gradient analyses (Figures 4 and 5) supports the claim that these models capture perceptually relevant signal components, aligning with biological expectations of visual processing.

However, the broader claim that these models enhance "visual decoding" in general is less substantiated, as the evaluation focuses solely on image retrieval tasks without exploring other decoding paradigms, such as image reconstruction. This limitation weakens the generalizability of the claim beyond retrieval. The evidence is clear and convincing within the scope of retrieval, but the paper overstates its implications for brain decoding as a whole without broader task validation.

**Essential References Not Discussed:**

N/A

**Experimental Designs Or Analyses:**

I reviewed the experimental designs in Sections 4 and 5, focusing on the retrieval tasks. The design is sound: training on averaged EEG/MEG repetitions, using a 90/10 train/validation split, and testing on unseen images with multiple seeds ensures reproducibility and reduces overfitting. The use of paired T-tests is statistically valid, with significance levels (p<0.05) appropriately reported. The usage of architectures and main training loss for alignment is mostly well-reported and ensures reproducibility. However, the method for creating a human-aligned model, though referenced, were not explained clearly in the paper.

**Methods And Evaluation Criteria:**

The methods—using pre-trained human-aligned image encoders (e.g., Dreamsim) and existing EEG/MEG encoders (e.g., NICE)—are appropriate for the retrieval task, leveraging established contrastive learning frameworks (InfoNCE loss) to align brain and image embeddings. The evaluation criteria, primarily top-1 and top-5 retrieval accuracy on the Things EEG2 and MEG datasets, are standard for retrieval tasks and align with prior work (e.g., Song et al., 2024). However, the reliance on a single EEG dataset (Things EEG2) and one MEG dataset limits the robustness of the findings across diverse brain signal recordings. While the datasets are well-suited for the task, the lack of variety in datasets and tasks (e.g., reconstruction) restricts the method's applicability.

**Other Comments Or Suggestions:**

N/A

**Other Strengths And Weaknesses:**

Strengths: The paper’s strength lies in its thorough experimentation—spanning multiple encoders, modalities, and participants—demonstrating robustness within the retrieval task. The idea of leveraging human-aligned models is conceptually sound, addressing the semantic collapse in text-paired contrastive learning (e.g., CLIP), and the biological insights (Section 5.5) add depth. The clarity of presentation, with figures (e.g., Figure 5) and tables, enhances accessibility.

Weaknesses: Originality is limited, as the core methods (Dreamsim, NICE) are borrowed, and the contribution is an application rather than a novel algorithmic advance. The significance is tempered by the narrow focus on retrieval, omitting tasks like reconstruction, which are critical for general brain decoding (e.g., Scotti et al., 2023, used Stable Diffusion for reconstruction). The reliance on a single EEG dataset (Things EEG2) further limits generalizability. While the direction is promising, the work feels accumulative, reinforcing the utility of human-aligned models rather than breaking new ground.

**Questions For Authors:**

Why was image reconstruction not explored alongside retrieval to support the broader claim of improved visual decoding? A response demonstrating feasibility or plans for reconstruction could strengthen the paper’s significance, potentially shifting my recommendation to weak accept. Without this, the claim feels overstated.

Can you justify the exclusive use of the Things EEG2 dataset and MEG dataset, given other image-paired neural datasets (e.g., Allen et al., 2022) could test robustness?

**Relation To Broader Scientific Literature:**

The paper mainly builds upon the Dreamsim work (Fu, 2023) to create its main results, proving its effectiveness over classical vision models like CLIP (Radford, 2021). In terms of decoding the paper is related to Decoding natural images from eeg for object recognition (Song 2024), which takes a step further by utilizing human-aligned image encoders for contrastive learning.

**Theoretical Claims:**

The paper does not present formal theoretical proofs but proposes a hypothesis: human-aligned encoders improve decoding by better capturing perceptual attributes reflected in brain signals. This is supported empirically.

---

> ### Author Rebuttal · Authors · 2025-03-28
>
> We thank reviewer ewMG for their thorough review and constructive feedback. Below, we provide our response to your questions as well as some clarifications on the points you raised.
>
> **Regarding the image reconstruction task:**
>
>  Thank you for this thoughtful comment. We agree that visual decoding goes beyond retrieval, and incorporating reconstruction would strengthen our claims. We will revise the framing to more clearly reflect our focus on evaluating whether human-aligned image embeddings improve alignment with brain activity. Since retrieval directly reflects this alignment, it was our primary evaluation metric.
>
> State-of-the-art reconstruction methods typically use pretrained diffusion models based on latents from the CLIP representation space and project brain activity to that using an alignment loss such as contrastive or MSE loss (Scotti et al. 2024, Li et al. 2024, Benchetrit et al. 2024). However, a challenge is that current generative models are trained on original CLIP embeddings, so using human-aligned embeddings would require retraining or fine-tuning the generative model. This task is broader than brain-image alignment and primarily tests the general utility of human-aligned embeddings as a latent space for generative diffusion models. We thought our setup provided a more direct and easier-to-evaluate scenario that allows us to isolate the effect of human alignment better, and that is why it was our focus.
>
> An alternative, inspired by MindEye, decouples retrieval and reconstruction. In our case, we can have two separate representation spaces: In one, brain signals are aligned with human-aligned embeddings for retrieval and with the original CLIP embeddings for reconstruction. We are also experimenting with a two-stage pipeline: brain signals are first mapped to the human-aligned space (for retrieval) and then further projected into the original CLIP space for reconstruction. This framework is implemented and under active evaluation. Preliminary results are promising, but further experimentation is needed to draw firm conclusions.
>
> We hope our ongoing work and future directions help address your concern regarding the broader impact of the study.
>
> **Regarding the use of other datasets:**
>
> Thank you for this question. Our primary focus is on EEG, as decoding performance in this modality remains relatively low and presents clear opportunities for improvement. We included MEG due to its similar input structure, allowing us to use the same architectures and demonstrate consistent gains from aligning neural signals with human-aligned image embeddings.
>
> We agree that including other modalities, such as fMRI, could further strengthen our claims. In response to your suggestion, we extended our experiments to the NSD dataset (Allen et al., 2022) and observed similar improvements in retrieval performance using human-aligned embeddings. These results, will be included in the revision and reinforce the generalizability of our findings across neural recording modalities.
>
> |       | CLIP     |          | DINOv2   |          | DINO     |          | ENSEMBLE |          | OpenCLIP |          | SynCLR   |          |
> |-------|----------|----------|----------|----------|----------|----------|----------|----------|----------|----------|----------|----------|
> |       | HA       | Base     | HA       | Base     | HA       | Base     | HA       | Base     | HA       | Base     | HA       | Base     |
> | top-1 | **45.0±4.3** | 23.8±2.4 | **45.3±3.6** | 22.5±2.1 | **49.1±3.7** | 45.9±3.7 | **54.6±4.3** | 48.1±4.0 | **49.5±4.3** | 35.4±3.8 | **58.2±4.6** | 48.2±4.3 |
> | top-5 | **76.6±4.2** | 53.8±4.1 | **75.5±3.5** | 49.1±3.2 | **79.7±3.7** | 76.8±3.6 | **83.4±3.7** | 79.6±3.4 | **79.2±4.1** | 67.8±4.3 | **86.7±3.6** | 78.7±3.1 |

---

### Decision · Program_Chairs · 2025-05-01

**Decision:**

Accept (poster)

**Comment:**

The paper presents an application human-aligned image encoders to enhance the decoding of visual information from brain signals, alongside a series of evaluations on EEG and MEG data. Reviewers reported felt that the  claim—that human-aligned image encoders improve visual decoding from brain signals in the form of image retrieval—is well supported, but there were some concerns that the claim could be strengthened by inlcuding additional modalities, such as fMRI data, and that the claim was perhaps too broadly articulated in the paper, given that there were no image reconstruction tasks reported. More specifically, reviewers reported the following strengths:
* The paper is well-written and easy to follow.
* Extensive experiments across various EEG and image encoders, demonstrate the effectiveness of the use of human-aligned image encoders.
* The proposed method consistently improved measured alignment across multiple models and subjects, in some cases by a wide margin.
* The work included interesting additional findings, including those related to spectral and temporal responses.

Specific limitations and concerns discussed by reviewers include:
* The paper focuses on image retrieval task, while image reconstruction from brain signals, is another relevant, and more challenging, task. Experiments on the reconstruction task would provide a more convincing demonstration of the method’s effectiveness. Nonetheless, the authors robustly defend their focus on retrieval as it provides a direct measure of alignment between brain activity and image embeddings.
* The novelty is limited. Although effective, the model’s design and alignment strategy draws heavily on prior work.

The authors are encouraged to revise their paper along the lines they discuss in reviewer responses, including re-framing the motivation to more clearly reflect a focus on image retrieval, and adding the additional results on fMRI data from (Allen et al., 2022).

While some reviewers increased their scores on rebuttal, no one reviewer definitively championed the paper (all recommending "weak accept"). I have reflected this in my recommendation.